# Wipi3 is essential for alternative autophagy and its loss causes neurodegeneration

Hirofumi Yamaguchi[1], Shinya Honda [1], Satoru Torii [1], Kimiko Shimizu [2], Kaoru Katoh[3], Koichi Miyake[4], Noriko Miyake[4], Nobuhiro Fujikake[1], Hajime Tajima Sakurai[1], Satoko Arakawa[1 ✉] & Shigeomi Shimizu [1 ✉]

Alternative autophagy is an Atg5/Atg7-independent type of autophagy that contributes to various physiological events. We here identify Wipi3 as a molecule essential for alternative autophagy, but which plays minor roles in canonical autophagy. Wipi3 binds to Golgi membranes and is required for the generation of isolation membranes. We establish neuron-specific Wipi3-deficient mice, which show behavioral defects, mainly as a result of cerebellar neuronal loss. The accumulation of iron and ceruloplasmin is also found in the neuronal cells. These abnormalities are suppressed by the expression of Dram1, which is another crucial molecule for alternative autophagy. Although Atg7-deficient mice show similar phenotypes to Wipi3-deficient mice, electron microscopic analysis shows that they have completely different subcellular morphologies, including the morphology of organelles. Furthermore, most Atg7/Wipi3 double-deficient mice are embryonic lethal, indicating that Wipi3 functions to maintain neuronal cells via mechanisms different from those of canonical autophagy.

[1] Department of Pathological Cell Biology, Medical Research Institute, Tokyo Medical and Dental University, 1-5-45 Yushima, Bunkyo-ku, Tokyo 113-8510, Japan. [2] Department of Biological Sciences, School of Science, The University of Tokyo, Tokyo 113-0033, Japan. [3] Biomedical Research Institute, National Institute of Advanced Industrial Science and Technology (AIST), 1-1-1 Higashi, Tsukuba, Ibaraki 305-8566, Japan. [4] Department of Biochemistry and Molecular Biology, Nippon Medical School, 1-1-5 Sendagi, Bunkyo-ku, Tokyo 113-8602, Japan. ✉email: arako.pcb@mri.tmd.ac.jp; shimizu.pcb@mri.tmd.ac.jp

Autophagy is a cellular process that degrades various intracellular components, including proteins and organelles[1,2]. Autophagy is morphologically characterized by double-membrane autophagosomes and single-membrane autolysosomes which degrade subcellular constituents using lysosomal enzymes[1,2]. Autophagic membranes originate from the endoplasmic reticulum (ER) membrane, particularly mitochondria-associated ER membranes (MAM), and the formation of autophagosomes is driven by functional complexes, including Unc-51-like autophagy activating kinase 1 (Ulk1) complexes, phosphatidylinositol 3-phosphate kinases (PI3Ks), autophagy-related (Atg) 9 complexes, the Atg5 conjugation system, and the microtubule-associated protein light chain 3 (LC3) conjugation system[1,2]. After their generation, autophagosomes fuse with lysosomes to become autolysosomes by a mechanism dependent on syntaxin17[3]. Previously, this mechanism was considered to work in a manner entirely dependent on Atg5, but recent observations have suggested that it functions even in the absence of Atg5[4].

In addition to this canonical autophagy, several types of non-canonical autophagy have been reported[5–8]. Atg5-independent alternative macroautophagy (hereafter described as alternative autophagy) is one such type of autophagy, in which cargos are engulfed by the *trans*-Golgi membrane[5], instead of by MAM elongation, and are degraded in autolysosomes. This mechanism has also been called the Golgi-membrane-associated degradation pathway (GOMED)[9]. GOMED/alternative autophagy uses the same molecules as canonical autophagy, such as Ulk1 and PI3K, at the initial step[5]. However, the core machinery is completely different, and Rab9 and Dram1, but not Atg9, the Atg5 conjugation system, and the LC3 conjugation system, are utilized in alternative autophagy[5,9,10].

A variety of potential physiological functions of canonical autophagy have been demonstrated by the analysis of systemic and tissue-specific *Atg*-gene knockout mice, including resistance to early neonatal starvation, clearance of neuronal protein aggregates, and maintenance of cardiac function[11]. On the other hand, the biological roles of alternative autophagy have not been fully clarified to date. Alternative autophagy is known to preferentially degrade undelivered proteins when protein trafficking from the Golgi to the plasma membrane (PM) is disrupted. Therefore, it has a physiological function to regulate the efficiency of secretion[9]. One example is alternative autophagy in glucose-deprived β-cells, in which the secretion of insulin granules is blocked under specific conditions to avoid a further reduction in blood glucose levels, and undelivered insulin is degraded by GOMED/alternative autophagy[9]. Alternative autophagy also plays a role in the elimination of mitochondria from erythrocytes[12] and dedifferentiating iPS cells[13], and in the protection of intestinal epithelial cells[14], heart muscle cells[15], and neurons[16] against various stressors.

A deeper understanding of the biological relevance of alternative autophagy is expected to be achieved from the analyses of knockout mice with targeted deletions in the genes specific to the alternative autophagic pathway. Therefore, to address this issue, we searched for molecules required for alternative autophagy, and identified Wipi3 as such a molecule. Wipi3 has been reported to play a role in regulating canonical autophagy[17], but our findings showed that Wipi3 mainly contribute to alternative autophagy and plays only minor roles in canonical autophagy. We also analyzed the phenotypes of neuron-specific Wipi3-knockout mice, and identified an essential role of Wipi3 in the maintenance of neuronal cells. Wipi3 degrades ceruloplasmin, which is delivered from the Golgi to the PM, and prevents abnormal iron deposition in neuronal cells. Neurological defects were also observed in Atg7-deficient mice[18], but the pathogenesis involved

is different from that of Wipi3-knockout mice, because the subcellular morphologies of the two types of mice were completely different, and because Atg7/Wipi3 double-deficient mice showed much more severe phenotypes than single-knockout mice. From these results, we concluded that both Atg5/Atg7-dependent canonical autophagy and Wipi3-dependent alternative autophagy maintain neuronal cells in a healthy state via different mechanisms.

## Results

**Hsv2 is an essential molecule for alternative autophagy in yeast.** To identify molecules involved in alternative autophagy, we searched a database of synthetic lethal genes for canonical autophagy-associated genes in yeast, because the concomitant loss of the two types of autophagic machinery is expected to reduce cell viability. As a result, we identified *hsv2* to be such a gene, together with either *atg18* or *atg21*. Hsv2 is a member of the β-propellers that bind polyphosphoinositides (PROPPIN) family proteins, which have seven tryptophan-aspartic acid (WD)-repeat domains and two phosphoinositide (PIP)-binding sites[19,20] (Fig. 1a). Because amphotericin B1 (AmphoB) disrupts protein trafficking from the Golgi to the PM and induces the autophagic response in *atg5Δ* yeast cells[9], we analyzed the role of Hsv2 in AmphoB-induced alternative autophagy. As described previously[9], AmphoB treatment generated Golgi stacks (Fig. 1b), as well as autophagosome-like double-membrane compartments within the cytoplasm (Fig. 1c), and autophagic body-like single-membrane structures within vacuoles (Fig. 1d) in *atg5Δ* yeast cells. In *hsv2Δatg5Δ* cells, we observed no abnormalities in the untreated cells (Fig. 1e), and observed stacked Golgi membranes in the cells treated with AmphoB (Fig. 1f). However, these Golgi membranes were abnormally swollen (Fig. 1f), and we never observed autophagosome-like or autophagic body-like structures (Fig. 1g). Quantitative analysis confirmed these findings (Fig. 1h), indicating that Hsv2 is crucial for the generation of autophagic structures, probably by affecting the Golgi membrane. Of note, morphological analysis was performed using *atg5Δ* cells lacking Pep4, a vacuolar protease, to avoid the degradation of autophagic body-like structures inside vacuoles[9]. Consistently, from the aspect of proteolysis, AmphoB-induced degradation of green fluorescent protein (GFP)-fused pho8Δ60 (a cytosolic protein) was observed in *atg5Δ* yeast cells, but not in *hsv2Δatg5Δ* cells (Fig. 1i), confirming that Hsv2 is essential for alternative autophagy in yeast.

**Wipi3 is a crucial molecule for alternative autophagy in mammals.** The mammalian orthologues of yeast Hsv2 are Wipi3 and Wipi4, and the former is slightly more similar to Hsv2 (Fig. 1a). Therefore, we deleted Wipi3 from Atg5-deficient (*Atg5^KO*) mouse embryonic fibroblasts (MEFs) using the CRISPR/Cas9 system (referred to as *Atg5^KO/Wipi3^Cr* MEFs) (Supplementary Fig. 1a, b) and induced alternative autophagy by the addition of etoposide, a DNA-damaging reagent. As described previously[5], the ultrastructural analysis demonstrated the etoposide-induced formation of autophagosomes (double-membrane structures) and autolysosomes (single-membrane structures digesting subcellular constituents) in *Atg5^KO* MEFs (Fig. 2a, Supplementary Fig. 2a). In contrast, such autophagic structures were not observed in etoposide-treated *Atg5^KO/Wipi3^Cr* MEFs (Fig. 2b, Supplementary Fig. 2b), and the exogenous expression of Wipi3 (Supplementary Fig. 1b) restored the induction of autophagic structures (Fig. 2c, Supplementary Fig. 2c). These data suggested that Wipi3 is required for the induction of etoposide-induced alternative autophagy in mammals, as observed for Hsv2 in yeast cells (Fig. 1).

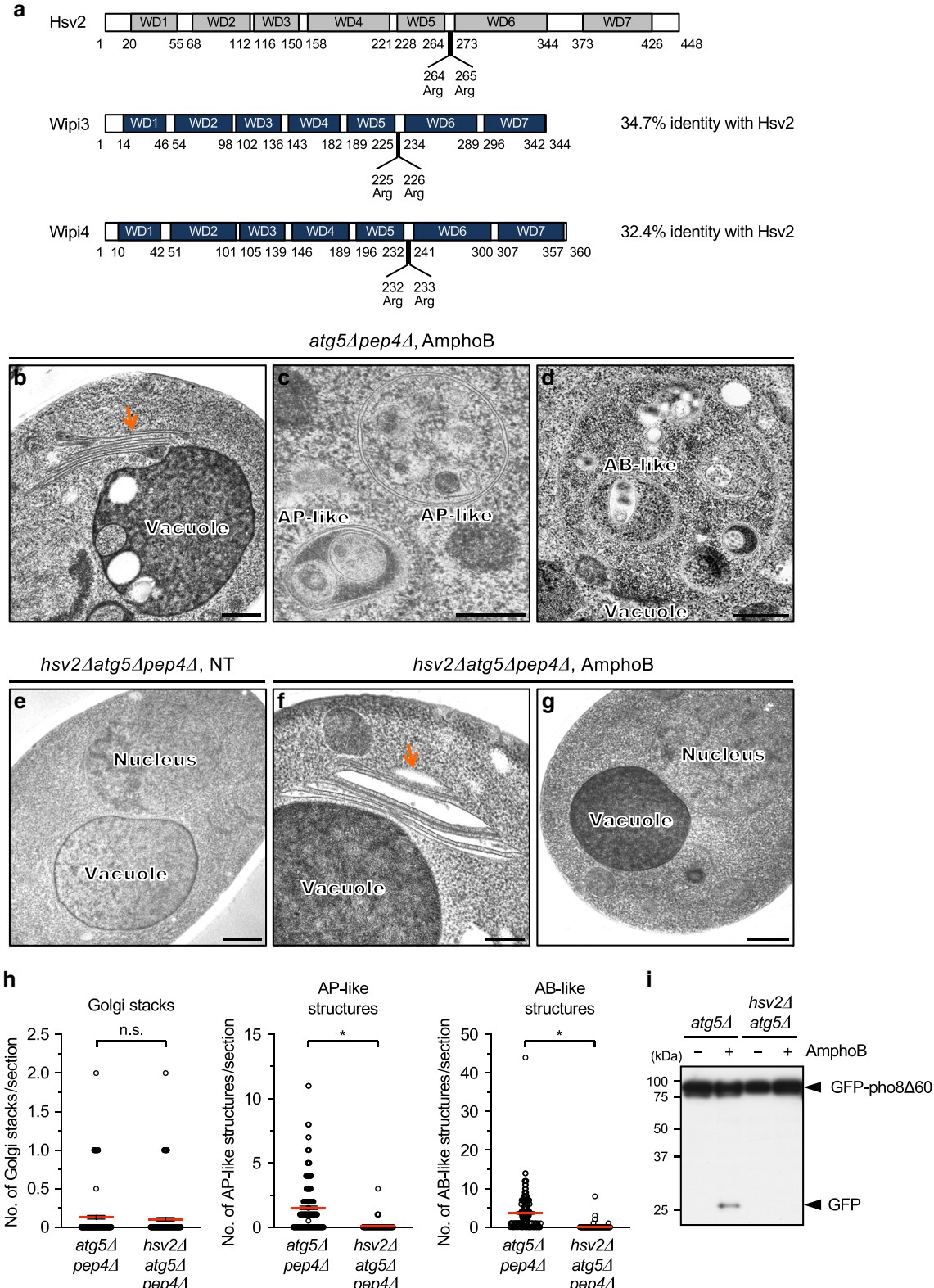

**h**

Golgi stacks
n.s.

AP-like
structures
*

AB-like
structures
*

**i**

To confirm the involvement of Wipi3 in alternative autophagy, we performed the monomer red-fluorescent protein (mRFP)-GFP tandem protein assay, which is widely used to detect autolysosomes, because GFP fluorescence, but not mRFP fluorescence, becomes weak within acidic compartments[21]. Correlative light and electron microscopy (CLEM) analysis confirmed that the red puncta were identical to autolysosomes (Supplementary Fig. 3a). Furthermore, Ulk1 and Ulk2 are molecules required for etoposide-induced alternative autophagy as well as canonical autophagy, and red puncta were not generated in $Atg5^{Cr}/Ulk1^{KO}/Ulk2^{KO}$ MEFs (Supplementary Fig. 4a–c), confirming the correct detection of autolysosomes. When this tandem protein was expressed within

**Fig. 1 Hsv2 is a crucial molecule for alternative autophagic responses in yeast. a** Structure of Hsv2 and its murine homologues Wipi3 and Wipi4. Seven WD domains and PIP-interacting sites (two arginine residues) are indicated. Amino acid numbers are shown below each protein. **b–g** Requirement of Hsv2 for the AmphoB-induced alternative autophagic response. $atg5\Delta pep4\Delta$ yeast cells (**b–d**) and $hsv2\Delta atg5\Delta pep4\Delta$ yeast cells (**e–g**) were not (NT) or were treated with AmphoB (2.5 μg/mL for 24 h) and observed using electron microscopy (EM). In $atg5\Delta pep4\Delta$ cells, Golgi stacking (**b**, arrow), autophagosome (AP)-like structures (**c**) and autophagic body (AB)-like structures (**d**) were observed. In $hsv2\Delta atg5\Delta pep4\Delta$ cells, stacked and swollen Golgi membranes (arrow) were observed (**f**). AP-like and AB-like structures were not observed in the cytosol and vacuoles, respectively (**g**). Bars = 0.5 μm (**b**, **d–g**) and 0.2 μm (**c**). **h** The number of Golgi stacks, and AP-like and AB-like structures per section are shown as the mean ± SEM ($atg5\Delta pep4\Delta$: n > 191 cells; $hsv2\Delta atg5\Delta pep4\Delta$: n > 100 cells). Red bars indicate mean values. Comparisons were performed using unpaired two-tailed Student t-tests. Left panel: ns: not significant ($p = 0.4137$). Other panels: *$p < 0.01$ (exact p values cannot be described since the value is too small [$p < 0.0001$]). **i** GFP–pho8Δ60-expressing $atg5\Delta$ cells and $hsv2\Delta atg5\Delta$ cells were treated with or without AmphoB (2.5 μg/mL for 24 h), and subjected to western blotting. The generation of free GFP is a marker of the autophagic response. Source data are provided as a Source Data file.

the cytoplasm, only a few red puncta were observed in nontreated $Atg5^{KO}$ MEFs, whereas etoposide-treated $Atg5^{KO}$ MEFs demonstrated many large red puncta that colocalized with the lysosomal protein Lamp2 (Fig. 2d). Red puncta were frequently observed in $Atg5^{KO}$ MEFs and Wipi3-expressing $Atg5^{KO}/Wipi3^{Cr}$ MEFs, but not in $Atg5^{KO}/Wipi3^{Cr}$ MEFs (Fig. 2d, Supplementary Fig. 3b). Quantitative analysis of cells with more than one red puncta (>1 μm) confirmed these findings (Fig. 2e), showing the essential role of Wipi3 in etoposide-induced alternative autophagy.

Autophagy can be more simply assessed by the formation of large Lamp2 puncta, as shown in Fig. 2d "Lamp2", because most red puncta from mRFP-GFP are included into the large Lamp2 puncta (Fig. 2d "merge"). The identity of the large Lamp2 puncta has been confirmed to be autolysosomes by CLEM analysis[5,10]. Consistent with the results of the mRFP-GFP assay, the large Lamp2 puncta assay showed the induction of alternative autophagy in $Atg5^{KO}$ MEFs and Wipi3-expressing $Atg5^{KO}/Wipi3^{Cr}$ MEFs, but not in $Atg5^{KO}/Wipi3^{Cr}$ MEFs upon etoposide treatment (Supplementary Fig. 5).

**Wipi3 is crucial for alternative autophagy-dependent proteolysis.** We also analyzed whether Wipi3 contributes to alternative autophagy-dependent proteolysis. To this end, we analyzed the degradation of the mCherry-Rab9 fusion protein, because we previously showed the existence of Rab9 on autophagic vacuoles[5]. When we treated mCherry-Rab9-expressing $Atg5^{KO}$ MEFs with etoposide, cleavage of mCherry-Rab9 and its inhibition by bafilomycin A1, an inhibitor of autolysosomal degradation, was observed (Fig. 2f), demonstrating that mCherry-Rab9 is a substrate of alternative autophagy. Importantly, this cleavage was not observed in etoposide-treated $Atg5^{KO}/Wipi3^{Cr}$ MEFs, and was recovered by the expression of Wipi3 (Fig. 2g, h), indicating the important role of Wipi3 in alternative autophagy-dependent proteolysis. Because the Rab9-fusion protein is degraded in autolysosomes, we visualized this degradation using mRFP-GFP-fused Rab9. We found that mRFP-GFP-Rab9 was localized in the cytoplasm as small yellow puncta in untreated $Atg5^{KO}$ MEFs (Fig. 2i: NT), which became big red puncta (owing to the autolysosomal quenching of the GFP fluorescence) surrounded by Lamp2 immunofluorescence after etoposide treatment (Fig. 2i: etoposide). Such puncta did not appear in etoposide-treated $Atg5^{KO}/Wipi3^{Cr}$ MEFs (Fig. 2j), indicating the Wipi3-dependent engulfment of mRFP-GFP-Rab9 into autolysosomes. The essential role of Wipi3 in alternative autophagy was confirmed by the treatment of $Atg7^{KO}/Wipi3^{KO}$ MEFs with etoposide (Supplementary Fig. 6), and when we used a different alternative autophagy inducer, 1,3-cyclohexanebis (methylamine), which is an inhibitor of Golgi trafficking[9,22] (Supplementary Fig. 7). All these data indicated that Wipi3 is required for alternative autophagy.

Note that apoptosis was induced by etoposide[23] concomitantly with alternative autophagy, but Wipi3 showed minimal effect on

apoptosis (Supplementary Fig. 8a, b). Furthermore, the autophagic response was not affected by the inhibition of apoptosis with the pan-caspase inhibitor q-VD-OPh (Supplementary Fig. 8c–f).

**Wipi3 localizes on the Golgi membrane upon etoposide treatment.** We next aimed to identify the step in alternative autophagy at which Wipi3 functions. Detailed ultrastructural analysis showed many ministacked Golgi and isolation membrane-like structures in etoposide-treated $Atg5^{KO}$ MEFs at early timepoints (Fig. 3a, Supplementary Fig. 9a). The latter structures were closely associated with the ministacked Golgi (Fig. 3a: right panels). This morphology was the same as that of etoposide-treated wild-type (WT) MEFs (Supplementary Fig. 9b). In contrast, in $Atg5^{KO}/Wipi3^{Cr}$ MEFs, ministacked Golgi were abnormally swollen and rod-shaped (Fig. 3b, Supplementary Fig. 9c), no isolation membrane-like structures were observed, and the morphologies of the other organelles were almost normal. Consistent EM findings were observed in etoposide-treated $Atg7^{KO}/Wipi3^{KO}$ MEFs (Supplementary Fig. 9d), suggesting that Wipi3 functions at the Golgi membranes before the generation of isolation membranes.

Therefore, we next analyzed the positional association between Wipi3 and Golgi membranes. To this end, we analyzed the intracellular localization of Wipi3 by stimulated emission depletion (STED) microscopy using Flag-Wipi3-expressing $Atg5^{KO}/Wipi3^{Cr}$ MEFs. This was because we found that Flag-Wipi3 was able to rescue the loss of Wipi3, and there are no available Wipi3 antibodies that can be used for immunofluorescence analyses (Supplementary Fig. 10). Flag-Wipi3 was evenly distributed throughout the cytoplasm in untreated MEFs, whereas a portion of Flag-Wipi3 translocated to the trans-Golgi (defined by Golgi SNAREs with a size of 15 kilodaltons [GS15]) after etoposide treatment (Fig. 3c, Supplementary Fig. 11). To visualize the etoposide-induced translocation of Wipi3 to the trans-Golgi, we performed the close proximity (Duolink) assay[24], which produces a fluorescent signal when two molecules are localized within 40 nm of each other. As shown in Fig. 3d, the number of Duolink signals between Flag-Wipi3 and HA-GS15 was time-dependently increased. Quantitative analysis confirmed the increase in Duolink intensity per cell (Fig. 3e), and the population of Duolink signal-positive cells (Fig. 3f), indicating that Wipi3 is translocated from the cytosol to the trans-Golgi membranes upon etoposide treatment. This translocation is thought to be crucial for the generation of isolation membranes.

**Localization of Wipi3 on autophagic structures.** We also analyzed the localization of Wipi3 at a later stage of autophagy, by visualizing Wipi3 and autolysosomes (red puncta of mRFP-GFP). Interestingly, most Wipi3 puncta were well colocalized with autolysosomes (Fig. 3g), supporting the involvement of Wipi3 in alternative autophagy. The colocalization of Wipi3 and red

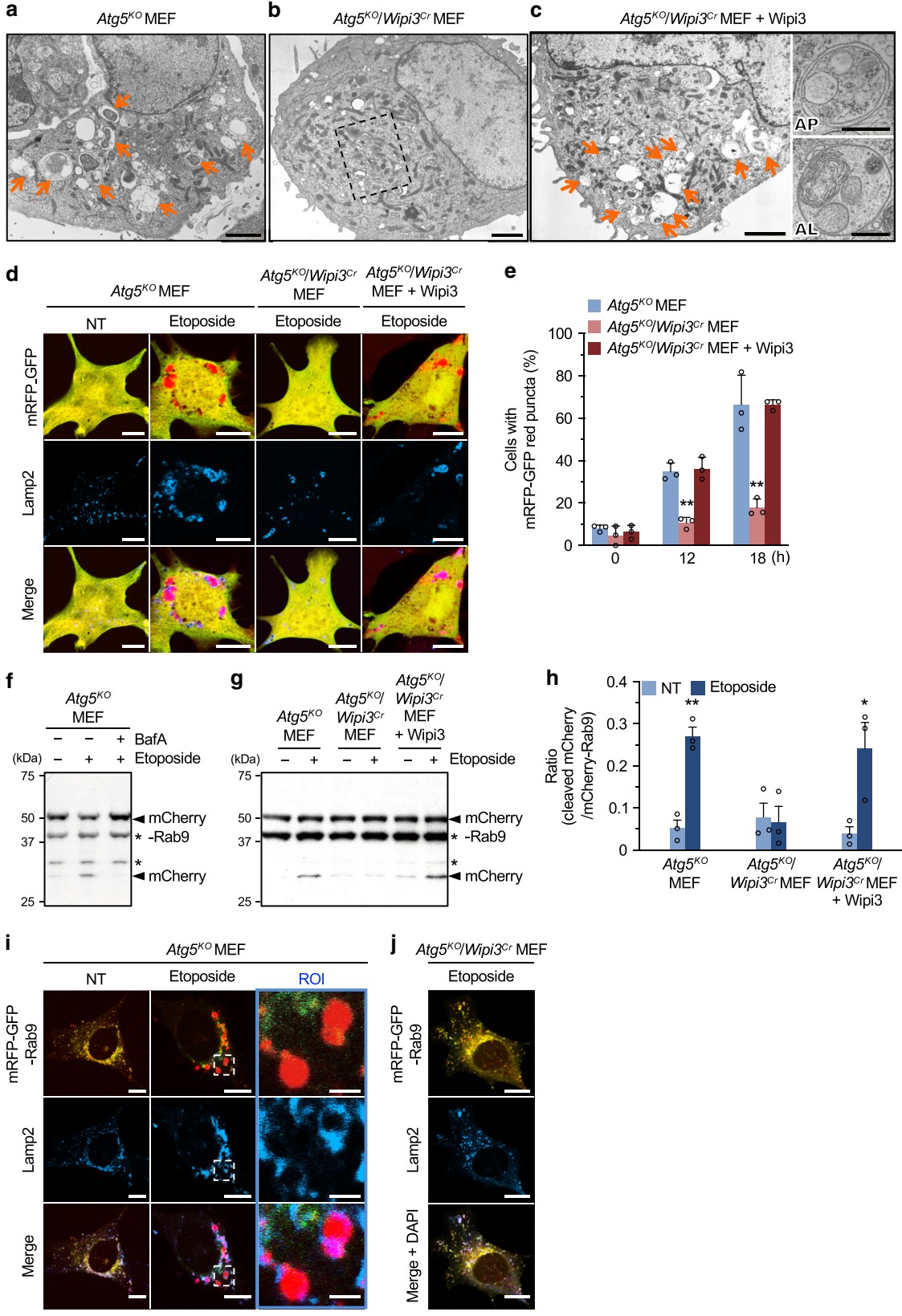

puncta is thought to be a result of Wipi3 being a component of autophagic membranes, but does not exclude the possibility that it is a cargo of autophagy. Therefore, we analyzed the nature of the isolated autophagic structures by immunoelectron microscopy. Upon lysis of the cells with isotonic buffer followed by sucrose-gradient centrifugation, both the *trans*-Golgi and

lysosomes were mainly recovered into fractions 4–6 (defined by GS15 and Lamp1) irrespective of etoposide treatment (Fig. 4a, Supplementary Fig. 12). Wipi3 levels increased in these fractions with longer times after etoposide treatment (Fig. 4a). EM analysis demonstrated that small round single-membrane vesicles were the major structure in fraction 5 of the untreated cells (Fig. 4b),

**Fig. 2 Wipi3 is essential for etoposide-induced alternative autophagy. a–c** Electron micrographs of the indicated MEFs treated with etoposide (10 μM) for 18 h. Arrows indicate autolysosomes. Bars = 2 μm. A magnified image of the dashed square is shown in Supplementary Fig. 2b. Quick freeze-substitution images of autophagosome (AP) and autolysosome (AL) are shown on the right. Bars = 0.5 μm. **d, e** The mRFP-GFP tandem protein assay showed the essential role of Wipi3 in alternative autophagy. The indicated MEFs expressing a mRFP-GFP protein were left untreated or were treated with etoposide (10 μM), and were immunostained with an anti-Lamp2 antibody. **d** Representative images at 18 h are shown. Bars = 10 μm. Red puncta indicate acidic compartments. **e** The populations of cells with red puncta (>1 μm) are shown (n > 30 cells). Data are shown as the mean ± SD (n = 3 experiments). **f–h** Degradation of mCherry-fused Rab9 demonstrates the essential role of Wipi3 in alternative autophagy. **f, g** The indicated MEFs expressing mCherry-Rab9 were treated without or with etoposide (10 μM; 18 h) and bafilomycin A1 (10 nM). The generation of free mCherry, a marker of the autophagic response, was analyzed by immunoblotting. Asterisks indicate nonspecific bands. **h** Cleavage efficiencies were obtained from western blots. Data are shown as the mean ± SEM (n = 3 experiments). **i, j** The indicated MEFs transiently expressing the mRFP-GFP-Rab9 protein were left untreated or were treated with etoposide (10 μM; 18 hr), and were immunostained with an anti-Lamp2 antibody. Red signals indicate mRFP-GFP-Rab9 proteins engulfed into acidic compartments. Bars = 10 μm. Magnified images are shown in the right panels. Bars = 2 μm. In (**e**), **\*\****p* < 0.01 vs the value of *Atg5*[KO] MEFs (12 h: *p* = 0.0020, 18 h: *p* < 0.0001). In (**h**), **\*\****p* < 0.01 vs the value of untreated *Atg5*[KO] MEFs (*p* = 0.0088) and *\*p* < 0.05 vs the value of untreated *Atg5*[KO]/ *Wipi3*[Cr] + *Wipi3* MEFs (*p* = 0.0243). Comparisons were performed using one-way ANOVA followed by the Tukey post hoc test. Source data are provided as a Source Data file.

whereas many autophagic structures, such as cup-shaped structures, double-membrane autophagosome-like structures, and autolysosome-like vacuoles containing multilamellar bodies were observed in the etoposide-treated cells (Fig. 4c). Such structures were rarely observed in the untreated cells (Fig. 4d). Importantly, immunoelectron microscopic analysis demonstrated the existence of Wipi3 and GS15 signals on the convex surface of the isolation membrane-like structures and autophagic vacuoles (Fig. 4e), confirming the localization of Wipi3 on Golgi-derived isolation membranes, and the subsequent formation of autophagic structures.

**Important role of PI3P in Wipi3 translocation to the Golgi membrane.** Wipi3 is known to interact with phosphatidylinositol 3-phosphate (PI3P), and two of its residues (R225 and R226) have been suggested to be crucial for this interaction[19,20] (Fig. 1a). We confirmed this finding using recombinant proteins (Supplementary Fig. 13a). Thus, we hypothesized that this interaction is crucial for Wipi3-mediated alternative autophagy. To investigate this point, we first analyzed the colocalization of Flag-Wipi3 and PI3P, which was visualized using mCherry-FYVE[25], in *Atg5*[KO]/ *Wipi3*[Cr] MEFs. Upon etoposide treatment, Wipi3 accumulated in the region next to the nucleus, where Golgi are localized, and PI3P also accumulated in the same area and merged well with the Wipi3 signals (Fig. 4f). Although Wipi3 also interacts with phosphatidylinositol 3,5-bisphosphate [PI(3, 5)P$_2$][19], we did not observe any colocalization of Wipi3 and PI(3, 5)P$_2$ (Supplementary Fig. 14). When we expressed a Flag-Wipi3 mutant (R225A or R226A) instead of Flag-Wipi3 at similar levels (Supplementary Fig. 1b), and treated the cells with etoposide, mutant Wipi3 proteins remained localized throughout the cytoplasm, without their colocalization with PI3P (Fig. 4f). Quantitative analysis confirmed the absence of the formation of Wipi3 puncta in mutant Wipi3-expressing cells (Supplementary Fig. 13b). Furthermore, STED analysis showed no translocation of Wipi3 mutants to the Golgi upon etoposide treatment (Supplementary Fig. 13c), indicating that the two residues (R225 and R226) are essential for the accumulation of Wipi3 to the Golgi. Importantly, the expression of Flag-Wipi3, but not its mutants, induced alternative autophagy in *Atg5*[KO]/*Wipi3*[Cr] MEFs, as assessed by the formation of large Lamp2 puncta formation (Fig. 4g, h). These data indicated the importance of the Arg[225] and Arg[226] residues of Wipi3 for its function in alternative autophagy, and also suggested the involvement of PI3P.

**Biological roles of Wipi3-dependent alternative autophagy.** We next aimed to clarify the biological roles of Wipi3-dependent

alternative autophagy. We previously demonstrated that alternative autophagy degrades proteins that are undelivered from the Golgi to the PM[9]. Therefore, we investigated the involvement of Wipi3 in this mechanism using vesicular stomatitis virus ts045 G protein fused to GFP (VSVG–GFP) (Fig. 5a). In *Atg5*[KO] MEFs, VSVG-GFP localizes to the ER at 40 °C (Fig. 5b, image 1), is transported to the Golgi at 15 min after a temperature shift to 32 °C (Fig. 5b, image 2), and is delivered to the PM by 60 min (Fig. 5b, image 3). Genotoxic stress, including etoposide, is known to alter Golgi morphology[26] and disrupt this delivery[27], and hence the amount of PM-localized VSVG-GFP was reduced (Fig. 5b, image 6). The decreased delivery of VSVG-GFP to the PM was confirmed by the staining of PM-localized VSVG-GFP followed by flow cytometry (Fig. 5c). Because undelivered VSVG-GFP molecules are thought to be degraded by alternative autophagy, we visualized these molecules by inhibiting substrate degradation using E64d and pepstatin, which are inhibitors of lysosomal proteases. Interestingly, a large amount of VSVG-GFP was enclosed by large Lamp2 puncta (Fig. 5d, image 3), indicating the autophagic degradation of VSVG-GFP in etoposide-treated *Atg5*[KO] MEFs. In marked contrast to *Atg5*[KO] MEFs, VSVG–GFP was observed not only in the Golgi but also in the cytoplasm as punctate structures in etoposide-treated *Atg5*[KO]/*Wipi3*[Cr] MEFs (Fig. 5b, images 12). The colocalization of VSVG-GFP and Lamp2 was not observed by the addition of E64d and pepstatin (Fig. 5d, image 6). Therefore, abnormal cytoplasmic puncta are thought to be VSVG-GFP molecules that have accumulated as a result of the lack of Wipi3-dependent alternative autophagy. Taken together, our results demonstrate that Wipi3 is important for the elimination of excess Golgi cargos upon genotoxic stress.

**Wipi3 and Wipi2 mainly contribute to alternative autophagy and canonical autophagy, respectively.** Because Wipi3 has also been reported to play a role in regulating canonical autophagy[17], we next compared the degree of its involvement with Wipi2, which is another member of the PROPPIN family and a molecule known to be involved in canonical autophagy[17,19,20,28]. As reported, starvation-induced canonical autophagy was substantially reduced in *Wipi2*[Cr] MEFs (Supplementary Fig. 15), but was much less reduced in *Wipi3*[Cr] MEFs (Fig. 6a–d), as assessed by LC3-II formation, p62 reduction, and LC3 puncta formation. Autophagic flux, calculated as the difference in LC3-II and p62 levels with or without bafilomycin A1, was also normal in *Wipi3*[Cr] MEFs (Fig. 6d). Similar results were observed when cells were treated with etoposide (Fig. 6e). Furthermore, colocalization analysis demonstrated that many Wipi2, but few Wipi3 molecules colocalized with LC3 in etoposide-treated WT MEFs (Fig. 6f), indicating the minimal involvement of Wipi3 in canonical

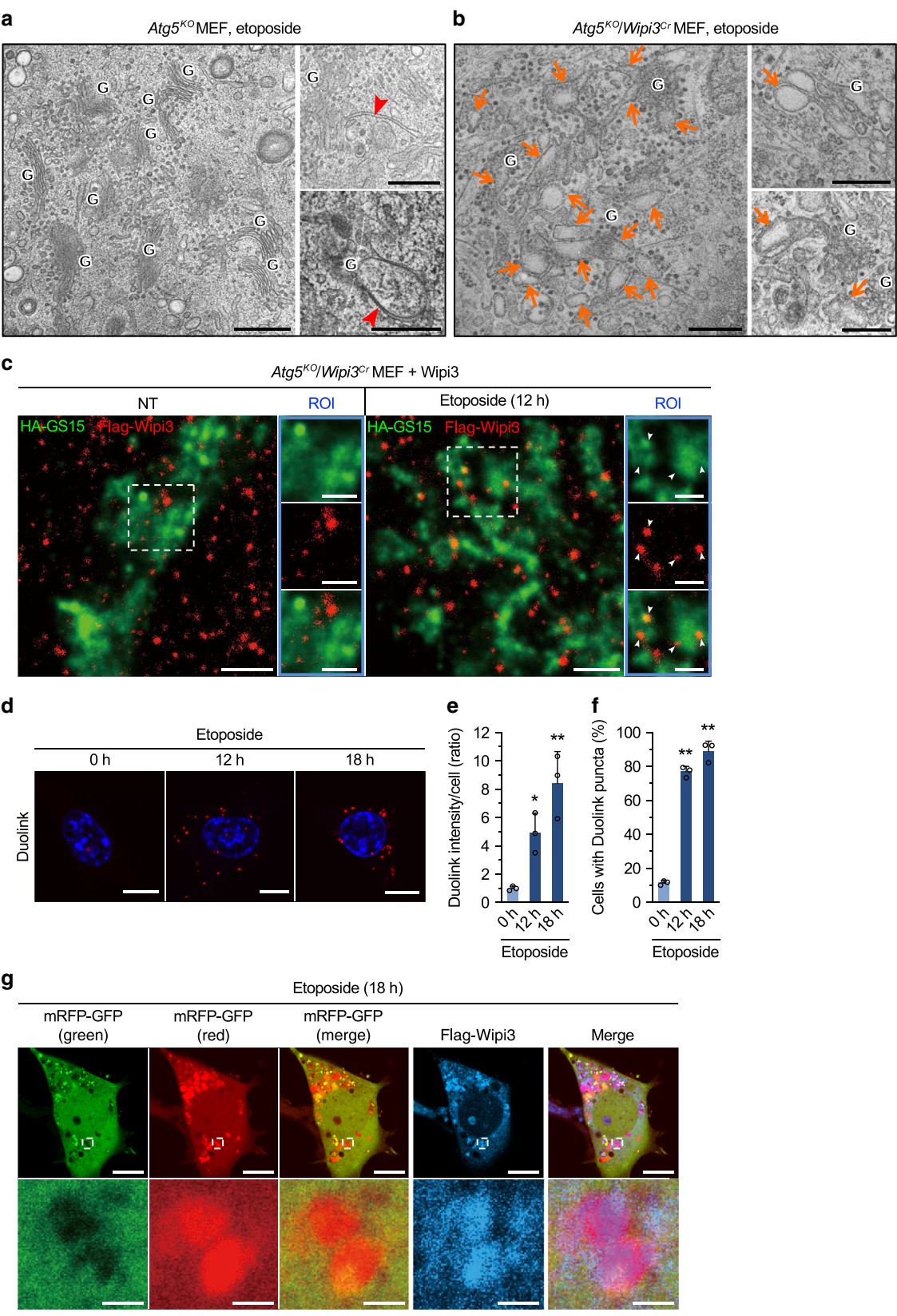

autophagy. Regarding the contribution of Wipi2 to alternative autophagy, we did not observe its etoposide-induced translocation to the Golgi (Supplementary Fig. 16a). Furthermore, when we treated $Atg5^{KO}/Wipi2^{Cr}$ MEFs with etoposide and compared them with $Atg5^{KO}$ MEFs, we did not find any differences as assessed by EM, the mRFP-GFP assay, the large Lamp2 puncta assay, and the mCherry-Rab9 cleavage assay (Supplementary Fig. 16b–g). Therefore, these data indicated that Wipi3 and Wipi2 contribute mainly to alternative autophagy and canonical autophagy, respectively.

**Fig. 3 Effects of Wipi3 on Golgi morphology in etoposide-treated $Atg5^{KO}$ MEFs. a, b** Electron micrographs of the indicated MEFs treated with etoposide (10 μM) for 12 hr. "G" indicates the Golgi apparatus, arrowheads indicate isolation membranes (**a**), and arrows indicate swollen rod-shaped Golgi membranes (**b**). Bars = 0.5 μm. **c** $Atg5^{KO}/Wipi3^{Cr}$ MEFs expressing Flag-Wipi3 and HA-GS15 were left untreated or were treated with etoposide (10 μM) for 12 h. Then, cells were immunostained and observed by STED. Red and green signals indicate Flag-Wipi3 and HA-GS15 (*trans*-Golgi), respectively. Bars = 1 μm. Magnified images are shown in the right panels. Bars = 0.5 μm. Arrowheads indicate Flag-Wipi3 localized on the *trans*-Golgi membrane. **d–f** The close proximity assay performed between Flag-Wipi3 and HA-GS15. $Atg5^{KO}/Wipi3^{Cr}$ MEFs were transiently transfected with Flag-Wipi3 and HA-GS15 for 24 h and treated with etoposide (10 μM). Then, the Wipi3–GS15 interaction was visualized using anti-Flag and anti-HA antibodies with a Duolink detection kit (**d**). Red signals indicate positive interactions. Bars = 10 μm. In (**e**), the signal intensities per cell were calculated and data are shown as proportions of the value of untreated cells (mean ± SD). In (**f**), the populations of MEFs with a positive Duolink signal were calculated. Data are shown as the mean ± SD ($n > 50$ cells examined over three independent experiments). **g** $Atg5^{KO}/Wipi3^{Cr}$ MEFs expressing mRFP-GFP and Flag-Wipi3 were treated with etoposide (10 μM) for 18 h. Then, cells were immunostained with an anti-Flag antibody. In the mRFP-GFP merged image, red puncta were well colocalized with Flag-Wipi3. Bars = 10 μm. Magnified images of the dashed squares are shown in the lower panels. Bars = 2 μm. In (**e**, **f**), comparisons were performed using one-way ANOVA followed by the Tukey post hoc test. In (**e**), *$p < 0.05$ and **$p < 0.01$ vs the value of 0 h (12 h: $p = 0.0473$, 18 h: $p = 0.0025$). In (**f**), **$p < 0.01$ vs the value of 0 h (Exact $p$ values cannot be described since the value is too small [$p < 0.0001$]). Source data are provided as a Source Data file.

**Crucial role of Wipi3 in alternative autophagy in WT MEFs.** As the contribution of Wipi3 to alternative autophagy was demonstrated using $Atg5^{KO}$ MEFs, we next analyzed whether Wipi3-dependent alternative autophagy would also be induced in WT MEFs. To this end, we analyzed the localization of autolysosomes (red puncta of mRFP-GFP) and Wipi3, and observed their colocalization in etoposide-treated WT MEFs (Fig. 6g). Combining these results with the fact that Wipi3 and LC3 puncta demonstrate different patterns in cells (Fig. 6f), alternative autophagy is thought to be induced in WT MEFs upon etoposide treatment. This notion was confirmed by ultrastructural analysis, in which characteristic Golgi morphology, i.e., swollen and rod-shaped Golgi membranes, was observed in $Wipi3^{Cr}$ MEFs and $Wipi3^{KO}$ MEFs upon etoposide treatment (Fig. 6h, i), which was the same as the morphology observed in etoposide-treated $Atg5^{KO}/Wipi3^{Cr}$ MEFs (Fig. 3b). Taken together, these results demonstrated that Wipi3 manipulates Golgi membranes and induces alternative autophagy in etoposide-treated WT MEFs.

**Loss of Wipi3 induces neuronal cell defects via a mechanism different from canonical autophagy.** To analyze the physiological role of alternative autophagy, we generated neuron-specific *wipi3*-conditional deficient ($Wipi3^{cKO}$) mice. Wipi3 proteins were absent from the brains of these mice but were present in control mice (Fig. 7a). $Wipi3^{cKO}$ mice demonstrated mild growth retardation from 8 weeks of age (Supplementary Fig. 17a, b), as well as motor defects, including abnormal limb-clasping reflexes (Fig. 7b) and abnormal footprint patterns (Fig. 7c). On the rotarod test, $Wipi3^{cKO}$ mice showed lower motor performance than littermate $Wipi3^{flox}$ mice (Fig. 7d). Neuropathological analyses showed severe cerebellar degeneration, i.e., immunostaining with an anti-calbindin antibody demonstrated marked loss of Purkinje cells in $Wipi3^{cKO}$ mice (Fig. 7e–g). Consistent with neuronal damage, we observed an increase in the glial marker glial fibrillary acidic protein (GFAP) (Fig. 7g) and the microglial marker Iba1 (Supplementary Fig. 17c). Neuronal damage was also observed in neurons of the cerebral cortex (Supplementary Fig. 17d), but the severity was much less than that observed in the cerebellum. In low-magnified EM images of $Wipi3^{cKO}$ mouse cerebella, we confirmed the presence of dead cells (Supplementary Fig. 18a), collapsed myelinated nerve fibers (Supplementary Fig. 18b), and lamellar bodies (Supplementary Fig. 18c); all of which are often observed in the brains of animals affected by neurodegenerative disorders, indicating that the loss of Wipi3 results in neurological defects. Importantly, we detected normal levels of canonical autophagy in the brains of $Wipi3^{cKO}$ mice, as assessed by their normal expression levels of Atg7, p62, and LC3-II on western blotting (Fig. 7a). Immunofluorescence analysis confirmed the

absence of p62 accumulation in the brains of $Wipi3^{cKO}$ mice (Fig. 7h).

Despite the similar neurological defects in $Wipi3^{cKO}$ mice and neuron-specific Atg7-deficient ($Atg7^{cKO}$) mice[18,29], their organellar morphologies were completely different. In the Purkinje cells of $Wipi3^{cKO}$ mice, we found the accumulation of small fragmented and swollen rod-shaped Golgi membranes (Fig. 7i, Supplementary Fig. 19a), which had a similar morphology to etoposide-treated $Wipi3^{Cr}$ MEFs and $Wipi3^{KO}$ MEFs (Fig. 6h, i). However, the Golgi apparatus in $Atg7^{cKO}$ Purkinje cells was intact (Fig. 7i, Supplementary Fig. 19a). Conversely, smooth ER membranes were highly enriched in $Atg7^{cKO}$ Purkinje cells, which might be owing to the blockage of ER membrane-derived canonical autophagy, but such ER alterations were not observed in $Wipi3^{cKO}$ Purkinje cells (Supplementary Fig. 19b). Regarding mitochondria, we observed swollen mitochondria in the Purkinje cells (Supplementary Fig. 20a), myelinated nerve fibers (Supplementary Fig. 20b), glomeruli (Supplementary Fig. 20c), and granular cells (Supplementary Fig. 20d) of $Wipi3^{cKO}$ cerebella, but not in WT and $Atg7^{cKO}$ cerebella (Supplementary Fig. 20) at early timepoints after onset. The presence of abnormal mitochondria suggested the disruption of mitochondrial elimination by alternative autophagy. Although we observed a small number of typical autophagosomes and autolysosomes in $Wipi3^{cKO}$ cerebella (Supplementary Fig. 21), these are thought to be structures involved in canonical autophagy.

**Accumulation of fibrils and ceruloplasmin in $Wipi3^{cKO}$ mice.** The loss of autophagy causes the abnormal accumulation of substances within the cytoplasm, which may lead to neurodegeneration. In $Atg7^{cKO}$ mouse cerebella, ubiquitinated proteins and large inclusion bodies were reported to be observed[18]; however, substantially fewer ubiquitinated proteins and inclusion bodies were observed in $Wipi3^{cKO}$ mouse cerebella (Fig. 7j, Supplementary Fig. 22). In contrast, we observed many fibrils and fibrils imperfectly enclosed by autophagic membrane-like structures in $Wipi3^{cKO}$ mouse cerebella (Fig. 7k, l, Supplementary Fig. 22), but much fewer of these structures were observed in $Atg7^{cKO}$ cerebella (Supplementary Fig. 22). Many myelinated nerve fibers of $Atg7^{cKO}$ and $Wipi3^{cKO}$ cerebella were filled with smooth ER and dense fibrils, respectively (Fig. 7m), suggesting that fibril accumulation was caused by the loss of Wipi3. Because neuronal fibrils could be originated from nuclear components, we suspected the possible involvement of nucleophagy in fibril accumulation[30]. However, analysis of DNA and nuclear factors did not show any apparent differences between Wipi3-expressing cells and Wipi3-deficient cells (Supplementary Fig. 23).

We further investigated the abnormally accumulated substances in $Wipi3^{cKO}$ mouse cerebella, and found the deposition of

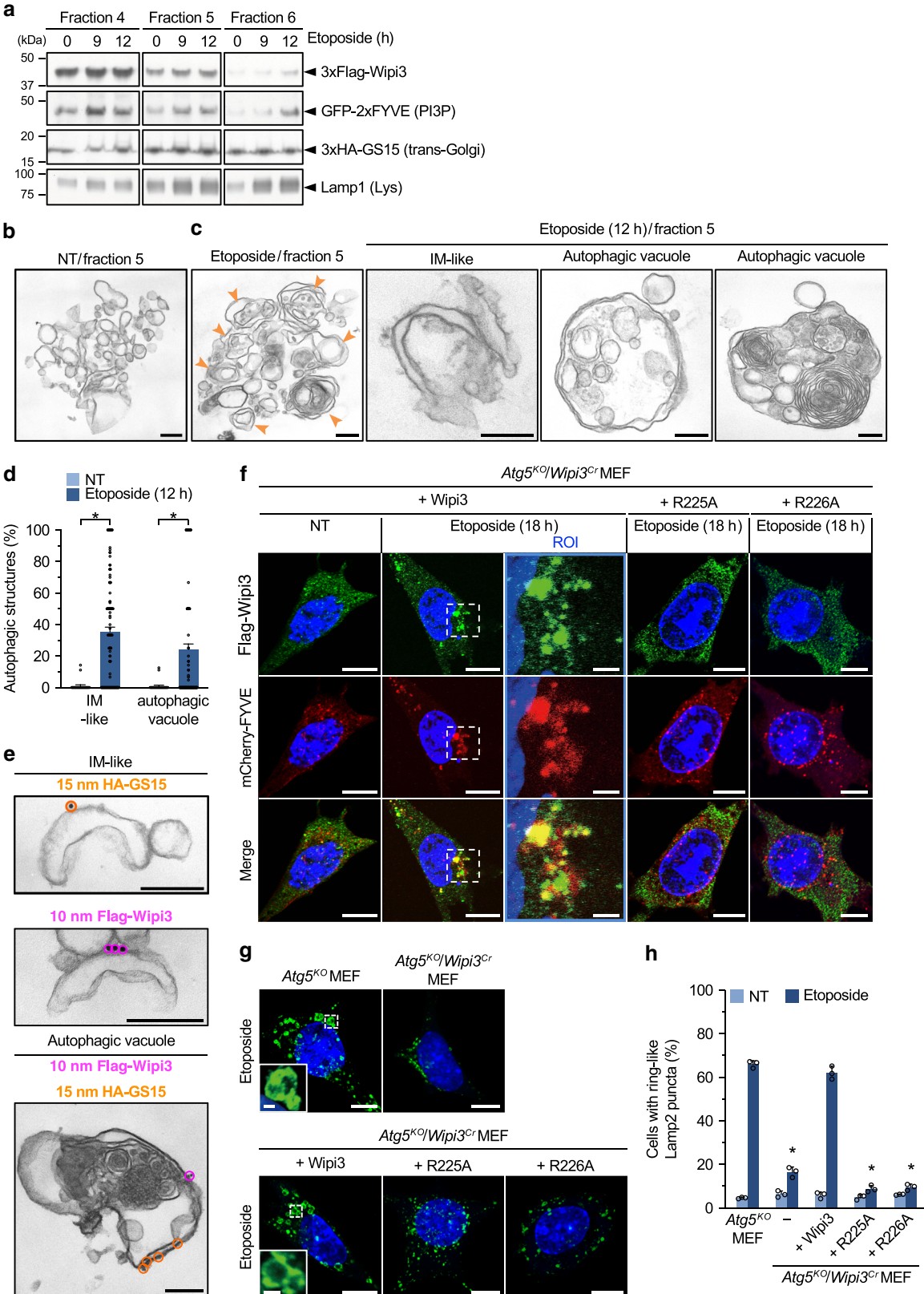

iron mainly in Purkinje and granular cells by Prussian blue staining, which detects $Fe^{3+}$ (Fig. 8a). We did not observe iron deposition in $Atg7^{cKO}$ mouse cerebella (Fig. 8a). Iron deposition is a possible disease trigger in various neurodegenerative disorders[31], and is associated with static encephalopathy of childhood with neurodegeneration in adulthood (SENDA) caused by *wipi4* gene mutations[32,33]. Aberrant iron deposition is usually associated with the abnormal accumulation of iron-binding proteins, and immunostaining analysis demonstrated the increased expression of ceruloplasmin, primarily in the Purkinje and granular cells of *Wipi3^{cKO}* mice (Fig. 8b). We also performed immunostaining for various other iron-binding proteins,

**Fig. 4 Localization of Wipi3 and GS15 on isolated autophagic membranes. a** Successful fractionation of *trans*-Golgi membranes. $Atg5^{KO}$ MEFs were treated with etoposide (10 μM), lysed in isotonic buffer, and fractionated by sucrose-gradient centrifugation. The expression of each molecules were analyzed by western blotting. Detailed data are shown in Supplementary Fig. 12. **b, c** Representative electron micrographs of fraction 5 from untreated and etoposide-treated $Atg5^{KO}$ MEFs. Arrowheads indicate autophagic vacuoles. Representative isolation membrane (IM)-like and autophagic structures are shown in the right panels. Bars = 0.2 μm. **d** The populations of IM-like and autophagic structures per total structures were calculated. Data are shown as the mean ± SEM (NT: $n = 25$ structures; Etoposide: $n = 122$ structures). **e** Immunoelectron micrographs of fraction 5 from etoposide-treated $Atg5^{KO}$ MEFs. HA-GS15 and Flag-Wipi3 were recognized by 15-nm gold-conjugated anti-HA antibodies and 10-nm gold-conjugated anti-Flag antibodies, respectively. Bars = 0.2 μm. **f** Association of Wipi3 with PI3P in etoposide-treated $Atg5^{KO}$ MEFs. $Atg5^{KO}/Wipi3^{Cr}$ MEFs expressing Flag-Wipi3 and its mutants together with mCherry-FYVE were left untreated or were treated with etoposide (10 μM). Green and red signals indicate Flag-Wipi3 derivatives and PI3P, respectively. Bars = 10 μm. Magnified images are shown in the right panels. Bars = 2 μm. **g, h** The indicated MEFs were treated with etoposide (10 μM; 18 h), and were stained with an anti-Lamp2 antibody. In (**g**), Bars = 10 μm. Magnified images are shown in the inset. Bars = 1 μm. Ring-like Lamp2 puncta were autolysosomes. **h** The populations of MEFs with ring-like Lamp2 puncta were calculated. Data are shown as the mean ± SD ($n > 200$ cells examined over 3 independent experiments). In (**d, h**), comparisons were performed using one-way ANOVA followed by the Tukey post hoc test. In (**d**), *$p < 0.01$ vs the value of untreated MEFs (IM-like: exact $p$ value is too small [$p < 0.0001$], autophagic vacuole: $p = 0.0083$). In (**h**), *$p < 0.01$ vs the value of etoposide-treated $Atg5^{KO}$ MEFs (exact $p$ value is too small [$p < 0.0001$]). Source data are provided as a Source Data file.

---

including ferritin, ferroportin, etc., but no abnormalities were observed (Supplementary Fig. 24). Ceruloplasmin is a copper-interacting enzyme that also contributes to iron metabolism as a ferroxidase, which oxidizes $Fe^{2+}$ into $Fe^{3+}$. Secreted ceruloplasmin is generated in the liver, but membrane-bound ceruloplasmin is expressed in multiple organs, including in the brain[34]. Importantly, membrane-bound ceruloplasmin is delivered from the Golgi to the PM, and as described above, molecules delivered from the Golgi are the target substrates of alternative autophagy. In fact, when we expressed ceruloplasmin in $Atg5^{KO}$ MEFs and treated them with etoposide in the presence of E64d and pepstatin (to visualize cargo in autolysosome), we observed large ring-like Lamp2 structures (autolysosomes) colocalized with Wipi3 and ceruloplasmin (Fig. 8c). In contrast, this colocalization was not observed in $Atg5^{KO}/Wipi3^{Cr}$ MEFs (Fig. 8d), indicating ceruloplasmin as a cargo of Wipi3-dependent alternative autophagy. Canonical autophagy was not involved in ceruloplasmin degradation, because we observed the colocalization of Wipi3, ceruloplasmin, and Lamp2, but not LC3, in WT MEFs (Fig. 8e), and because we did not observe ceruloplasmin accumulation in $Atg7^{cKO}$ Purkinje cells (Fig. 8b). Taken together, these results demonstrate that ceruloplasmin is degraded by Wipi3-dependent alternative autophagy, and hence ceruloplasmin was accumulated in $Wipi3^{cKO}$ mouse cerebella.

**Suppression of neurological defects in $Wipi3^{cKO}$ mice by Dram1.** To exclude the possibility that the phenotype of $Wipi3^{cKO}$ mice was caused by the loss of Wipi3 functions other than those in alternative autophagy, we injected lentiviruses containing Dram1, a molecule that mainly activates alternative autophagy[10], into the cerebella of $Wipi3^{cKO}$ mice at 5 weeks of age (Supplementay Fig. 25). Dram1 is a Golgi-localizing protein and its overexpression alone is able to induce alternative autophagy[10]. Importantly, the overexpression of Dram1 induced alternative autophagy even in $Atg5^{KO}/Wipi3^{Cr}$ MEFs (Supplementay Fig. 26). Mice were analyzed at 10 weeks of age, and broad transfection of the lentivirus throughout the cerebellum was confirmed by the expression of internal ribosome entry site (IRES)-mediated GFP (Fig. 9a). The introduction of Dram1 and Wipi3 (positive control), but not the vector (negative control) into the cerebellum of these mice rescued the abnormal neurological phenotypes, such as defects on the footprint test (Fig. 9b) and rotarod analyses (Fig. 9c). Consistently, Dram1 suppressed Purkinje cell loss (Fig. 9d), prevented iron deposition (Fig. 9e), and rescued intracellular abnormalities, including abnormal-shaped Golgi (Fig. 9f). Dram1 also suppressed the formation of collapsed myelinated nerve fibers (Fig. 9g, h). The ultrastructural analysis also showed several autophagic structures engulfing

fibrils in Dram1-transfected and Wipi3-transfected $Wipi3^{cKO}$ mouse cerebella (Supplementay Fig. 27). These structures were not observed in $Wipi3^{cKO}$ mouse cerebella (Fig. 7k, l). Because different alternative autophagy molecules were able to suppress the phenotype of $Wipi3^{cKO}$ mice, alternative autophagy is thought to play a role in neuronal cell maintenance. All of these results indicate that Wipi3 is an essential molecule for alternative autophagy, and protects neurons via mechanisms other than canonical autophagy.

**$Atg7^{cKO}/Wipi3^{cKO}$ mice showed a more severe phenotype than single-knockout mice.** Because the intracellular morphologies of $Wipi3^{cKO}$ and $Atg7^{cKO}$ neuronal cells were substantially different, their pathogenesis was thought to be different. To confirm this possibility, we generated neuron-specific $Atg7^{cKO}/Wipi3^{cKO}$ mice. Both $Atg7^{cKO}$ and $Wipi3^{cKO}$ mice were born in the expected Mendelian ratio, whereas most neuron-specific $Atg7^{cKO}/Wipi3^{cKO}$ mice were embryonic lethal (Fig. 10a, b). Furthermore, all the born mice demonstrated severe growth retardation (Fig. 10c), and rapidly died within 4 weeks (Fig. 10a, b). These mice showed motor defects (Fig. 10d) with severe cerebellar degeneration (Fig. 10e) at 4 weeks, which was earlier than that observed in $Atg7^{cKO}$ and $Wipi3^{cKO}$ mice. On EM analysis, we observed many dead Purkinje cells and dead granular cells (Supplementary Fig. 28). Furthermore, in viable neurons, we observed the accumulation of small fragmented and swollen rod-shaped Golgi membranes (Fig. 10f) with accumulated fibrils (Fig. 10g), and severely damaged myelinated nerve fibers (Fig. 10h), which had a similar morphology to $Wipi3^{cKO}$ mice at 10 weeks of age. We also observed highly enriched smooth ER membranes, as in $Atg7^{cKO}$ mice (Fig. 10i), indicating that the neuronal cells of $Atg7^{cKO}/Wipi3^{cKO}$ mice show combined morphological features of each single-knockout mouse. The acceleration of disease severity and the combined morphological features indicated a different pathogenesis between $Atg7^{cKO}$ and $Wipi3^{cKO}$ mice. Thus, both canonical and alternative autophagy are expected to protect neuronal cells via different mechanisms.

**Discussion**
The biological roles of alternative autophagy have not been well understood to date, because the molecules specific to alternative autophagy had not yet been identified. In this study, we identified Wipi3 as such a molecule. We also showed that Wipi3 is translocated from the cytosol to the *trans*-Golgi, and manipulates the *trans*-Golgi membrane to generate autophagic vacuoles upon etoposide treatment. In vivo, Wipi3-dependent alternative autophagy is required for the maintenance of neuronal cells via a

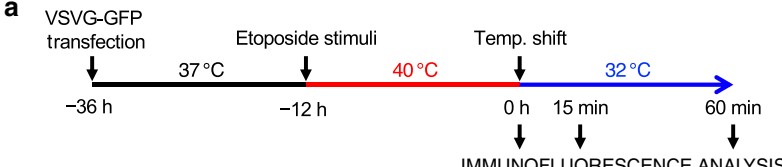

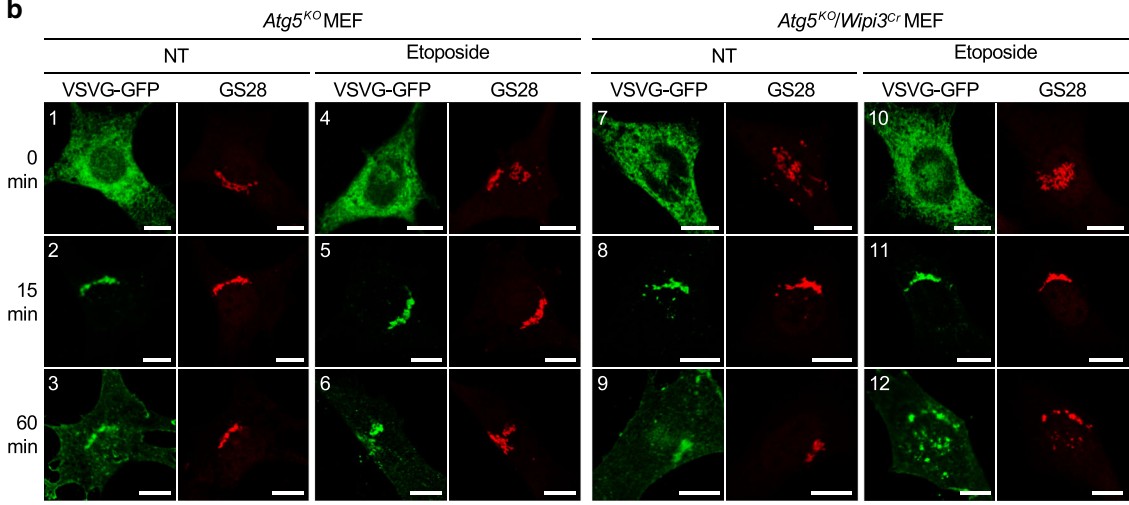

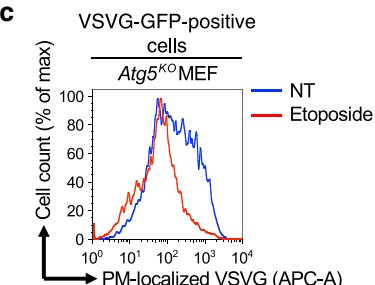

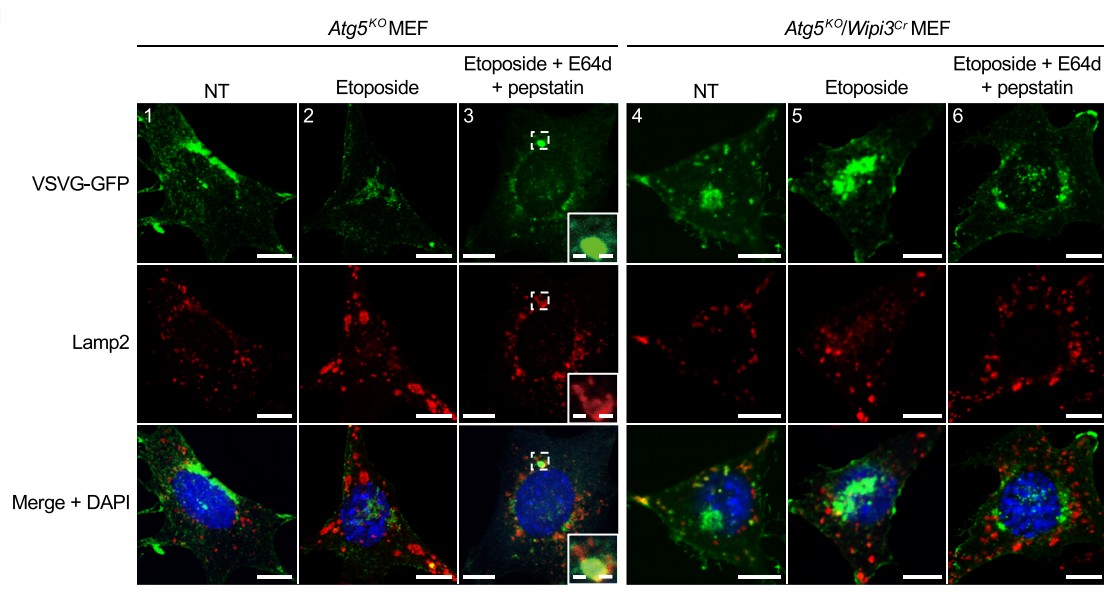

different mechanism to that of Atg7-dependent canonical autophagy.

We here analyzed the ultrastructural morphology of *Wipi3^{KO}*, *Wipi3^{Cr}*, *Atg5^{KO}/Wipi3^{Cr}*, and *Atg7^{cKO}/Wipi3^{KO}* MEFs and *Wipi3^{cKO}* and *Atg7^{cKO}/Wipi3^{cKO}* neuronal cells. The most notable and common alteration among these cells occurred in the Golgi, in which Golgi membranes were ministacked, swollen, and rod-shaped. Such alterations in Golgi membranes were not observed in *Atg5^{KO}* MEFs and *Atg7^{cKO}* neuronal cells, and have also not been reported in numerous EM analyses of canonical autophagy-deficient cells.

**Fig. 5 Effects of Wipi3 on VSVG trafficking upon etoposide treatment. a** Schematic diagram of the experiment. VSVG–GFP-expressing *Atg5KO* MEFs and *Atg5 KO/Wipi3Cr* MEFs were treated with or without etoposide (10 μM) at the restrictive temperature (40 °C) for 12 h. Cells were then shifted to the permissive temperature (32 °C), and fixed at the indicate times. **b** Representative images at 0, 15, and 60 min after the temperature shift are shown. Golgi were stained with an anti-GS28 antibody, and the location of VSVG-GFP and Golgi were analyzed. Scale bars = 10 μm. **c** Reduction of PM-localized VSVG–GFP upon etoposide treatment in *Atg5KO* MEFs. The *Atg5KO* MEFs containing VSVG-GFP were or were not treated with etoposide, and at 60 min after the temperature shift, PM-localized VSVG-GFP was stained with an anti-VSVG antibody, and its amount was measured using flow cytometry. **d** The indicated MEFs containing VSVG-GFP were treated with or without etoposide (10 μM) in the presence of E64d and pepstatin. At 60 min after the temperature shift, cells were immunostained with an anti-Lamp2 antibody. Bars = 10 μm. Magnified images of the dashed squares are shown in the insets. Bars = 1 μm.

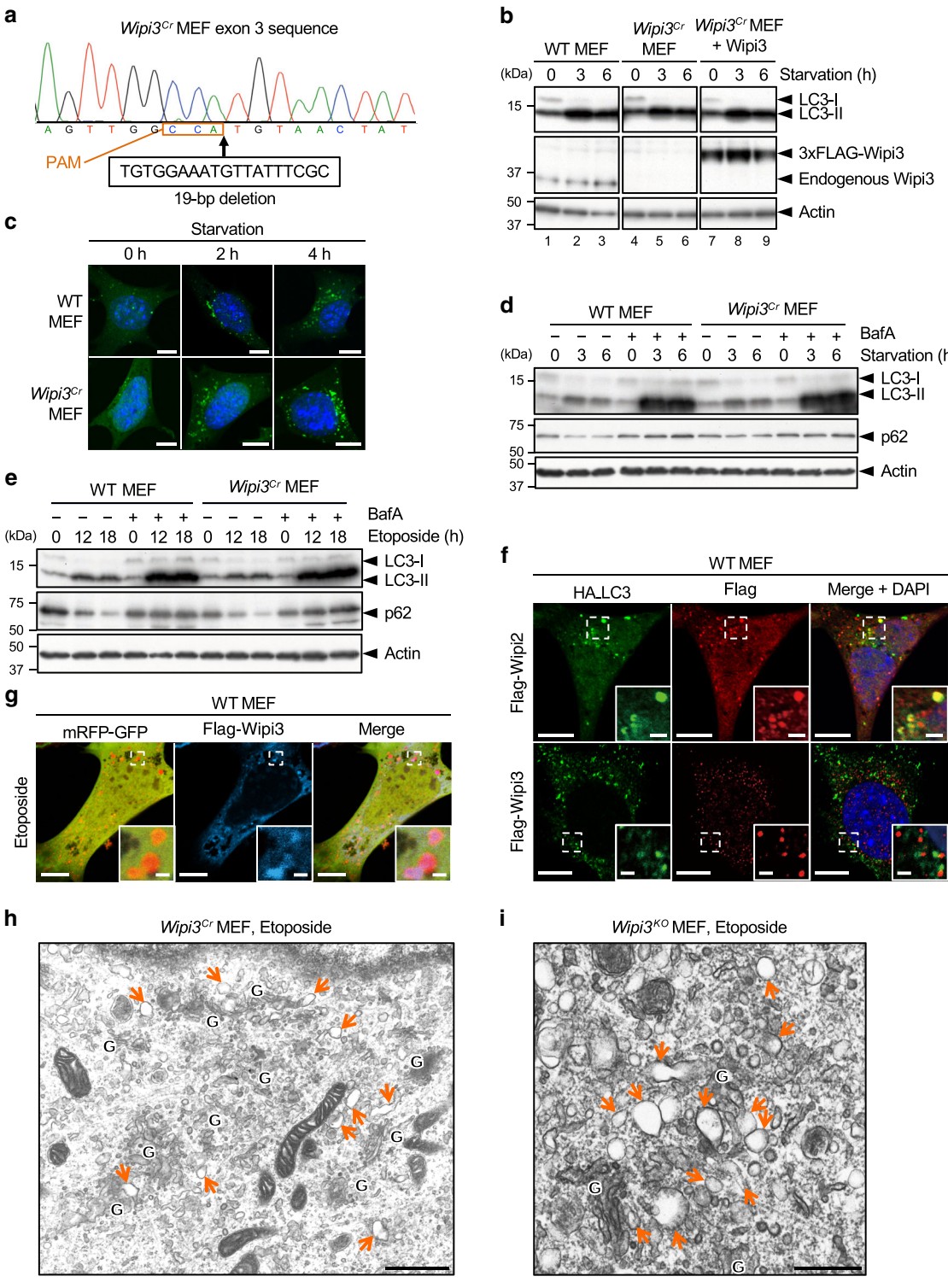

**Fig. 6 Comparison of the contribution of Wipi3 and Wipi2 to canonical autophagy. a** $Wipi3^{Cr}$ MEFs were generated from WT MEFs using the CRISPR/Cas9 system. A 19-bp deletion of *wipi3* (third exon) was confirmed by genomic sequencing. The deleted nucleotide sequence and the protospacer adjacent motif (PAM) are indicated below the sequence. **b** The indicated MEFs were starved for the indicated times, and the expression of the indicated proteins was analyzed by western blotting. Actin was included as a loading control. **c** The indicated MEFs expressing GFP-LC3 were starved for the indicated times, and GFP-LC3 puncta formation was analyzed. Representative images are shown. Bars = 10 μm. **d** The indicated MEFs were starved with or without bafilomycin A1 (10 nM) for the indicated times, and the expression of each protein was analyzed by western blotting. **e** Similar experiments to (**d**) were performed using etoposide (10 μM), instead of starvation. The deletion of *wipi3* showed a minimal effect on canonical autophagy. **f** Flag-Wipi2 or Flag-Wipi3 were expressed in HA-LC3-expressing WT MEFs, and were treated with etoposide (10 μM) for 12 h. Then, cells were immunostained with anti-HA and anti-Flag antibodies. Bars = 10 μm. Magnified images of the dashed squares are shown in the insets. Bars = 2 μm. LC3 signals were merged with Wipi2 but not with Wipi3. **g** Flag-Wipi3 and mRFP-GFP were expressed in WT MEFs, and were treated with etoposide (10 μM) for 12 h. Then, cells were immunostained with an anti-Flag antibody. Bars = 10 μm. Magnified images of the dashed squares are shown in the insets. Bars = 1 μm. Red puncta (autolysosomes) were well merged with Wipi3 signals. **h, i** Electron micrographs of $Wipi3^{Cr}$ MEFs (**h**) and $Wipi3^{KO}$ MEFs (**i**) treated with etoposide (10 μM) for 12 h. "G" indicates the Golgi apparatus, and arrows indicate swollen rod-shaped Golgi membranes. Bars = 1 μm (**h**) and 0.5 μm (**i**). Source data are provided as a Source Data file.

Therefore, Wipi3 is thought to affect to the Golgi membranes in a manner different from that of canonical autophagy. The morphologies of MEFs and neurons lacking Wipi3 were basically the same, but the accumulation of fibrils was only observed in Wipi3-deficient neurons. This is thought to be owing to differences in the cell types or the time required for of accumulation; i.e., many substrates are known to take a long time to accumulate in neurons. The loss of both Wipi3 and Atg5 or Atg7 additionally abolished a small fraction of autophagic structures from MEFs and neurons, and induced the accumulation of abnormal smooth ER in neurons, which was owing to the blockage of canonical autophagy.

Wipi3 has been reported to be a regulator of canonical autophagy[17]. However, our findings showed that Wipi2 and Wipi3 dominantly functions in canonical autophagy and alternative autophagy, respectively. Wipi2 is a paralog of Wipi3, with a similar structure, including the PI3P-interacting domain. However, the roles of Wipi2 and Wipi3 are different. The main cause of this difference is thought to be the efficiency of translocation to the Golgi upon a stimulus. We here described the requirement of the PI3P-interacting domain for translocation of Wipi3 to the Golgi. However, this domain is not sufficient for this translocation. Another crucial signal sequence is thought to be required for the translocation of Wipi3 to the Golgi, which may be absent in Wipi2.

Both canonical and alternative autophagy require the same molecules, such as Ulk1 and PI3K, at their initial step[1,2,5]. However, they use different molecules in the later steps. Therefore, it is likely that similar molecules are involved in the intermediate steps. In this context, the different involvement of Wipi3 and Wipi2 mainly in alternative and canonical autophagy, respectively, is quite reasonable. Wipi family members are reported to target membranes through nonspecific electrostatic curvature-dependent interactions, to be retained at the membranes through PIP binding[35–38], and to elongate membranes via their lipid-transfer activity[39–41]. In this context, Wipi2 (as well as Wipi1) bind PI3P and localize to the limiting membranes, such as nascent autophagosomal membranes, and function in the elongation of isolation membranes by recruiting Atg12-Atg5-Atg16 complexes[28,42]. Analogously, Wipi3 also interacts with *trans*-Golgi membranes using its PI3P interaction motif upon etoposide treatment, which may function in the elongation of isolation membranes from Golgi membranes for alternative autophagy. Alternatively, Wipi3 may alter the nature of Golgi membranes for their elongation.

Interestingly, neurological defects were observed in $Wipi3^{cKO}$ mice, which were similar to those of $Atg7^{cKO}$ mice[18,29]. The neuropathological analysis also demonstrated the degeneration of Purkinje neurons and reactive gliosis in both types of mice. However, the pathogenesis involved is thought to be different,

because (1) canonical autophagy was absent in $Atg7^{cKO}$ mice, but was present in $Wipi3^{cKO}$ mice (Fig. 7a, h), (2) polyubiquitinated proteins and inclusion bodies were abnormally accumulated in $Atg7^{cKO}$ mice but not in $Wipi3^{cKO}$ mice, whereas many fibrils as well as iron accumulation were observed in $Wipi3^{cKO}$ mice but not in $Atg7^{cKO}$ mice (Figs. 7 and 8), and (3) $Atg7^{cKO}/Wipi3^{cKO}$ mice showed much more severe phenotypes than single-knockout mice (Fig. 10). These findings showed that Wipi3 maintains neuronal cells in a healthy state via a different mechanism to that of canonical autophagy, i.e., probably via the digestion of different molecules.

How Wipi3 maintains neuronal cells in a healthy state remains unclear to date. One possible mechanism is the prevention of iron deposition, because $Wipi3^{cKO}$ mice showed neurodegeneration with iron accumulation, and iron deposition is often associated with various neurodegenerative diseases and can trigger or accelerate neuronal damage[31]. Although the precise pathogenesis occurring in $Wipi3^{cKO}$ mice remains unclear, a likely scenario is as follows: (1) ceruloplasmin is transported from the Golgi to the PM, and excess ceruloplasmin is degraded by alternative autophagy in healthy neuronal cells, (2) the ceruloplasmin is abnormally accumulated and mislocalized in the cytosol owing to the lack of its degradation in the $Wipi3^{cKO}$ neuronal cells, and (3) because ceruloplasmin has ferroxidase activity (from $Fe^{2+}$ to $Fe^{3+}$), mislocalized ceruloplasmin generates excess $Fe^{3+}$ molecules and deposits them in the cytosol. Alternatively, excess $Fe^{3+}$ may be released to the cell exterior, and binds with transferrin, resulting in the reuptake of $Fe^{3+}$ into neurons[34]. Because ceruloplasmin deficiency is also reported to induce iron deposition owing to a dysfunction in $Fe^{2+}$ to $Fe^{3+}$ conversion[43], an appropriate ceruloplasmin level is important for the maintenance of healthy neurons. In $Wipi3^{cKO}$ mice, multiple abnormalities, including abnormal iron storage and the accumulation of fibrils and damaged mitochondria, may participate in inducing neurodegeneration.

SENDA is a human neurodegenerative disorder characterized by iron deposition[32], and is caused by a *wipi4* gene mutations[33]. Our preliminary studies suggested that Wipi4 also functions in alternative autophagy, but to a smaller extent than Wipi3. Consistently, although *wipi4*-deficient mice developed a neurodegenerative disorder, the phenotypes were much milder than those of *wipi3*-deficient mice[44], and they did not show substantial iron accumulation[44]. Thus, the phenotypes of *wipi3*-deficient mice might be more close to human SENDA than those of *wipi4*-deficient mice. Regarding the *wipi3* gene mutations in human neurodegenerative diseases, they have been suggested to cause intellectual disability with development delay and progressive spastic quadriplegia[45,46]. In these people, there is substantial reduction of cerebral white matter, which is the same as that

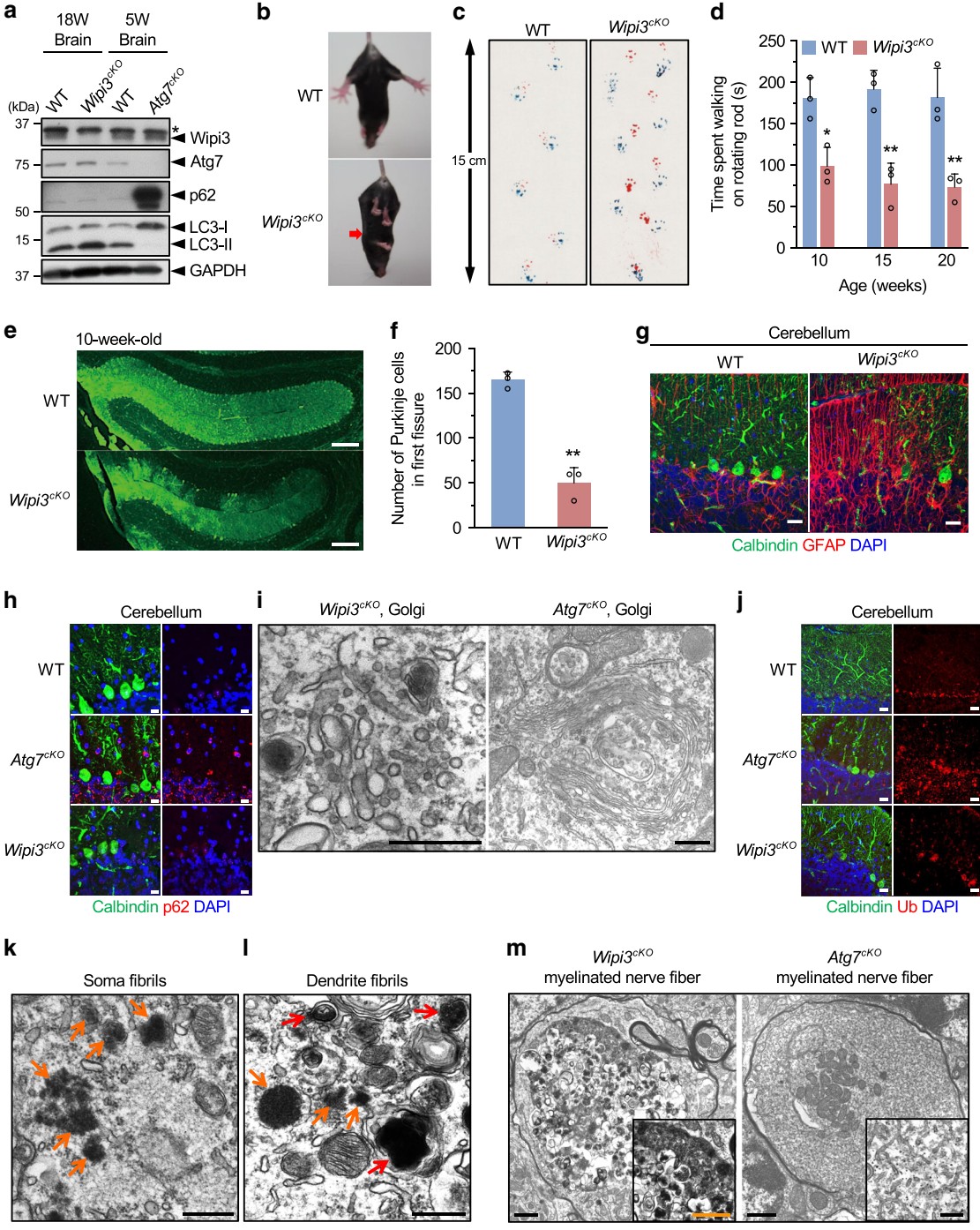

**Fig. 7 Neurological defects in neuron-specific *Wipi3cKO* mice. a** Western blot analysis of brain lysates. The asterisk indicates a nonspecific band. *Wipi3cKO* mice at 18-weeks were compared with *Atg7cKO* mice at 5-weeks, as they showed equivalent disease severity. **b–d** Abnormal motor performance in *Wipi3cKO* mice at 10-weeks. The limb-clasping reflex was observed (**b**). The footprint assay indicated a motor deficit (**c**). In (**d**), the time the indicated mice remained on the rod was measured. Data are shown as the mean ± SD (*n* = 3). **e–g** Cryosections of the indicated cerebellum were immunostained for the Purkinje cell marker calbindin (green), the glial cell marker GFAP (red), and DAPI (blue). Bars = 200 μm (**e**) and 20 μm (**g**). In (**f**), the number of Purkinje cells in the first fissure of the cerebellum is shown. Data are shown as the mean ± SD (*n* = 3). **h–m** Comparisons of morphology between WT, *Wipi3cKO* mice (10-weeks), and *Atg7cKO* mice (5-weeks). In (**h**), cryosections of the cerebellum immunostained with anti-calbindin and anti-p62 antibodies. Bar = 10 μm. (**i**) Morphology of Golgi in cerebella. Small fragmented and swollen rod-shaped Golgi membranes were observed in the Purkinje cells from *Wipi3cKO* mice. Bars = 0.5 μm. **j** Immunostaining of calbindin and ubiquitin (Ub) using cryosections of the cerebellum. Bars = 20 μm. **k**, **l** The accumulation of fibrils was observed in Purkinje cells from *Wipi3cKO* mice. Bars = 0.5 μm. Orange and red arrows indicate naked fibrils and fibrils imperfectly enclosed by autophagy-like membranes, respectively. (**m**) Myelinated nerve fibers of *Atg7cKO* cerebella and *Wipi3cKO* cerebella were filled with smooth ER and dense fibrils, respectively; bars = 1 μm (*Wipi3cKO*) and 0.5 μm (*Atg7cKO*). Magnified images are shown in the inset; bars = 1 μm (*Wipi3cKO*) and 0.2 μm (*Atg7cKO*). In (**c**), comparisons were performed using one-way ANOVA followed by the Tukey post hoc test. **p* < 0.05 and ***p* < 0.01 (10, 15, 20 weeks: *p* = 0.0175, 0.0013, 0.0021, respectively.) In (**f**), comparisons were performed using unpaired two-tailed Student *t*-tests. ***p* < 0.01 (*p* = 0.0005). Source data are provided as a Source Data file.

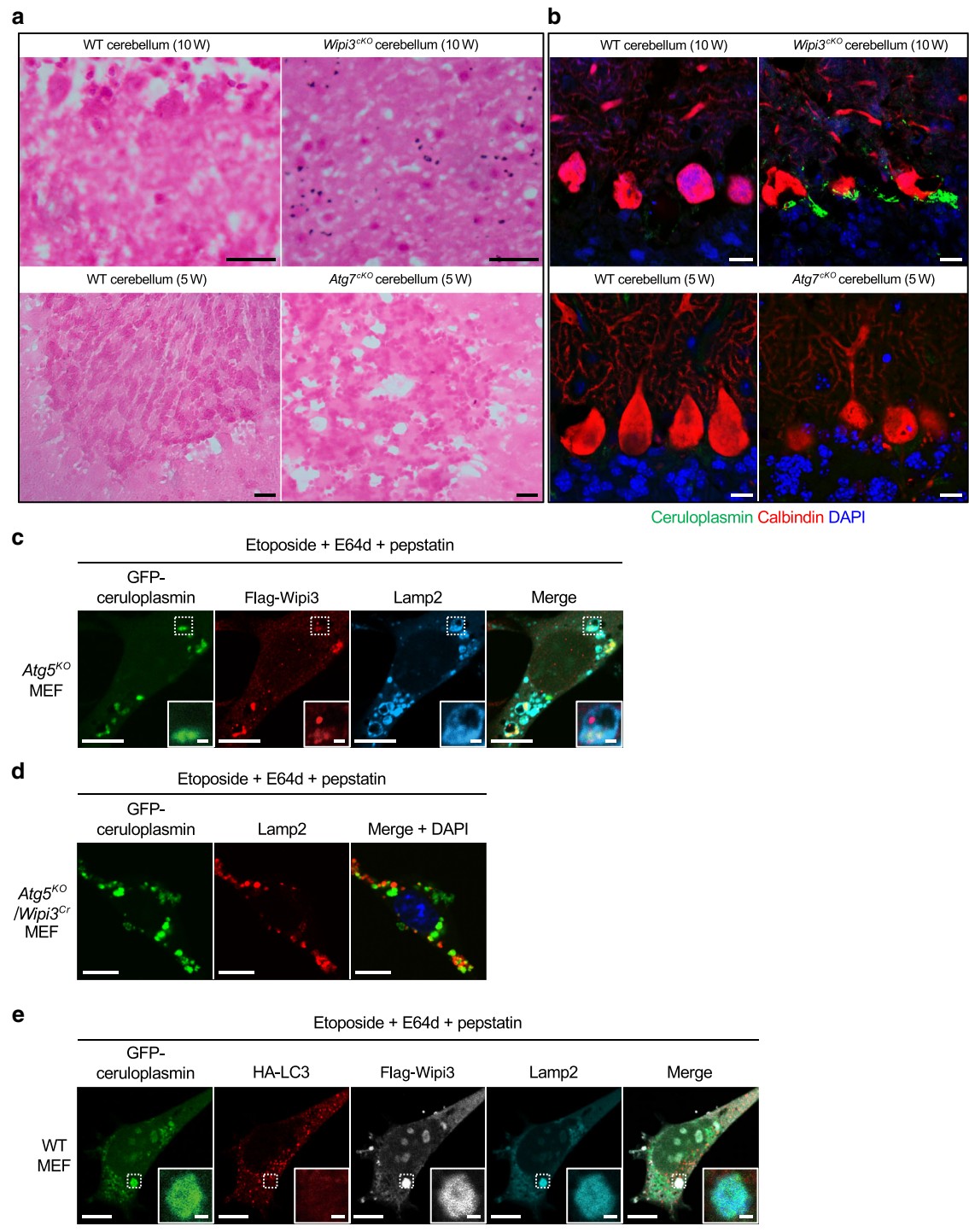

**Fig. 8 Accumulation of iron and ceruloplasmin in neuron-specific *Wipi3cKO* mice. a, b** Cryosections of the cerebellum from the indicated mice were stained with Prussian blue (**a**) and were immunostained with anti-ceruloplasmin (green) and anti-calbindin (red) antibodies (**b**). Bars = 20 μm (**a**) and 10 μm (**b**). Blue puncta indicate iron deposition in (**a**). (**c**) *Atg5KO* MEFs expressing GFP-ceruloplasmin and Flag-Wipi3 were treated with etoposide (10 μM) for 12 h in the presence of E64d and pepstatin. Then, cells were immunostained with anti-Flag (red) and anti-Lamp2 (blue) antibodies. Bars = 10 μm. Magnified images of the dashed squares are shown in the insets. Bars = 1 μm. Ceruloplasmin is incorporated into Wipi3-positive large Lamp2 puncta in *Atg5KO* MEFs. **d** A similar experiment to (**c**) was performed using *Atg5KO/Wipi3Cr* MEFs. Bars = 10 μm. **e** WT MEFs expressing GFP-ceruloplasmin, HA-LC3, and Flag-Wipi3 were treated with etoposide (10 μM) for 12 h in the presence of E64d and pepstatin. Then, cells were immunostained with anti-HA (red), anti-Flag (white), and anti-Lamp2 (blue) antibodies. Bars = 10 μm. Magnified images of the dashed squares are shown in the insets. Bars = 1 μm. LC3 is not colocalized with ceruloplasmin/Wipi3-positive large Lamp2 puncta.

observed in the neurons of *Wipi3cKO* mice. In mice, cerebellar damage is prominent, and hence *wipi3* mutations are expected to be identified as the cause of yet-uncharacterized cerebellar diseases. Taken together, we here identified Wipi3 as a molecule essential for alternative autophagy, and also showed its importance in the maintenance of neurons.

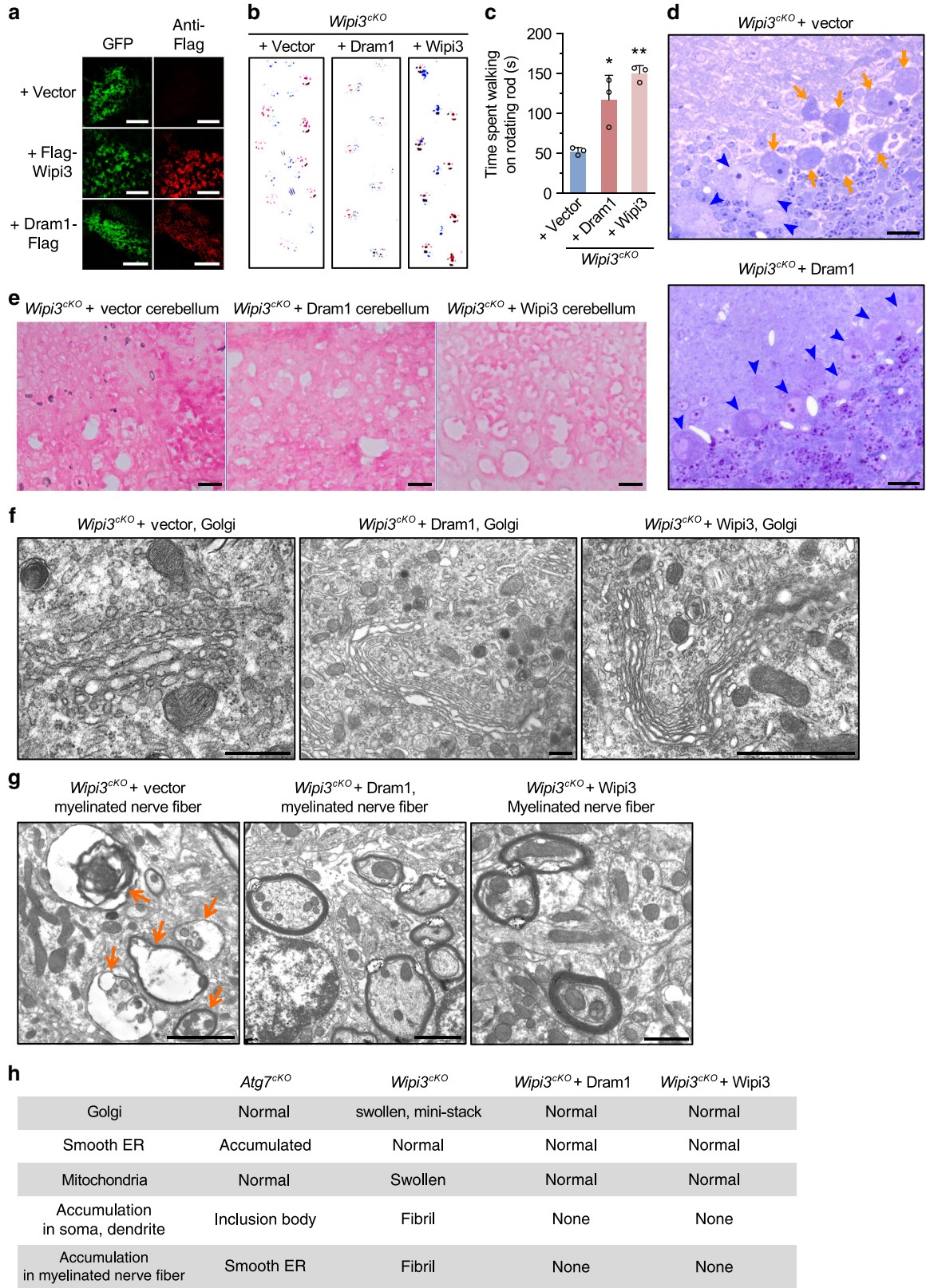

## Methods

**Reagents.** The following antibodies were used for the immunoblot and immunofluorescence assays: anti-GFP (Santa Cruz Biotechnology, #sc-9996), anti-Wipi3 (Thermo Fisher Scientific, #PA5-50864), anti-Atg5 (Sigma, #A0731), anti-Lamp2 (Abcam, #ab13524), anti-RFP (MBL, #M204-3), anti-Flag (MBL, #PM020), anti-HA (Santa Cruz Biotechnology, #sc-7392), anti-Lamp1 (abcam, #ab24170), anti-VSVG (KeraFAST, #EB0010), anti-GS28 (BD, #611184), anti-GM130 (BD, #610823), anti-Tom20 (Santa Cruz Biotechnology, #sc-11415), anti-Myc (Santa Cruz Biotechnology, #sc-40), anti-Rab9 (Sigma, #R5404), anti-His (Nacalai Tesque, #04428-26), anti-LC3 (nanoTools, #0231-100), anti-p62 (MBL, #PM045), anti-Wipi2 (abcam, #ab105459), anti-actin (Millipore, #MAB1501), anti-GST (Santa Cruz Biotechnology, #sc-138), anti-Atg7 (CST, #2631), anti-GAPDH (EMD Millipore, #MAB374), anti-calbindin D-28k (Swant, #300), anti-calbindin D-28k (Sigma, #C2724), anti-GFAP (Cell Signaling Technology #12389), anti-Iba1 (Wako, #019-19741), anti-ubiquitin (CST, #3933), anti-ceruloplasmin (BD, #611488), anti-ferritin (abcam, #ab69090), anti-ferroportin (Novus, #NBP1-21502)

**Fig. 9 Suppression of the neurological defects of *Wipi3cKO* mice by Dram1. a** The indicated genes were transfected with a lentivirus containing IRES-GFP into the cerebella of *Wipi3cKO* mice at 5-weeks of age (Supplementary Fig. 25). At 10-weeks, the expression of GFP and other molecules were confirmed by GFP fluorescence and immunostaining using an anti-Flag antibody. Bars = 200 μm. **b, c** Recovery of abnormal motor performance of *Wipi3cKO* mice by the expression of Dram1 at 10 weeks. Both the footprint assay (**b**) and rotarod test (**c**) demonstrated the suppression of the motor deficits of *Wipi3cKO* mice by Dram1. Data are shown as the mean ± SD (n = 3 experiments). **d** Semi-thin sections of the cerebellum from *Wipi3cKO* mice transfected with an empty vector or a vector for Dram1 were stained with toluidine blue. Orange arrows and blue arrowheads indicate dead and live Purkinje cells, respectively. Bars = 20 μm. Dram1 rescued Purkinje cell loss. **e** Cryosections of the cerebellum from the indicated mice were stained with Prussian blue. Bars = 20 μm. **f, g** Morphology of the Golgi (**f**) and myelinated nerve fibers (**g**) from the cerebella of the indicated mice. Swollen Golgi and damaged myelinated fibers were rescued by Dram1. Bars = 0.5 μm (**f**) and 5 μm (**g**). **h** Table summarizing the EM observations. In (**c**), comparisons were performed using one-way ANOVA followed by the Tukey post hoc test. *$p < 0.05$ and **$p < 0.01$ vs the value of Vector (Dram1: $p = 0.0135$, Wipi3: $p = 0.0017$). Source data are provided as a Source Data file.

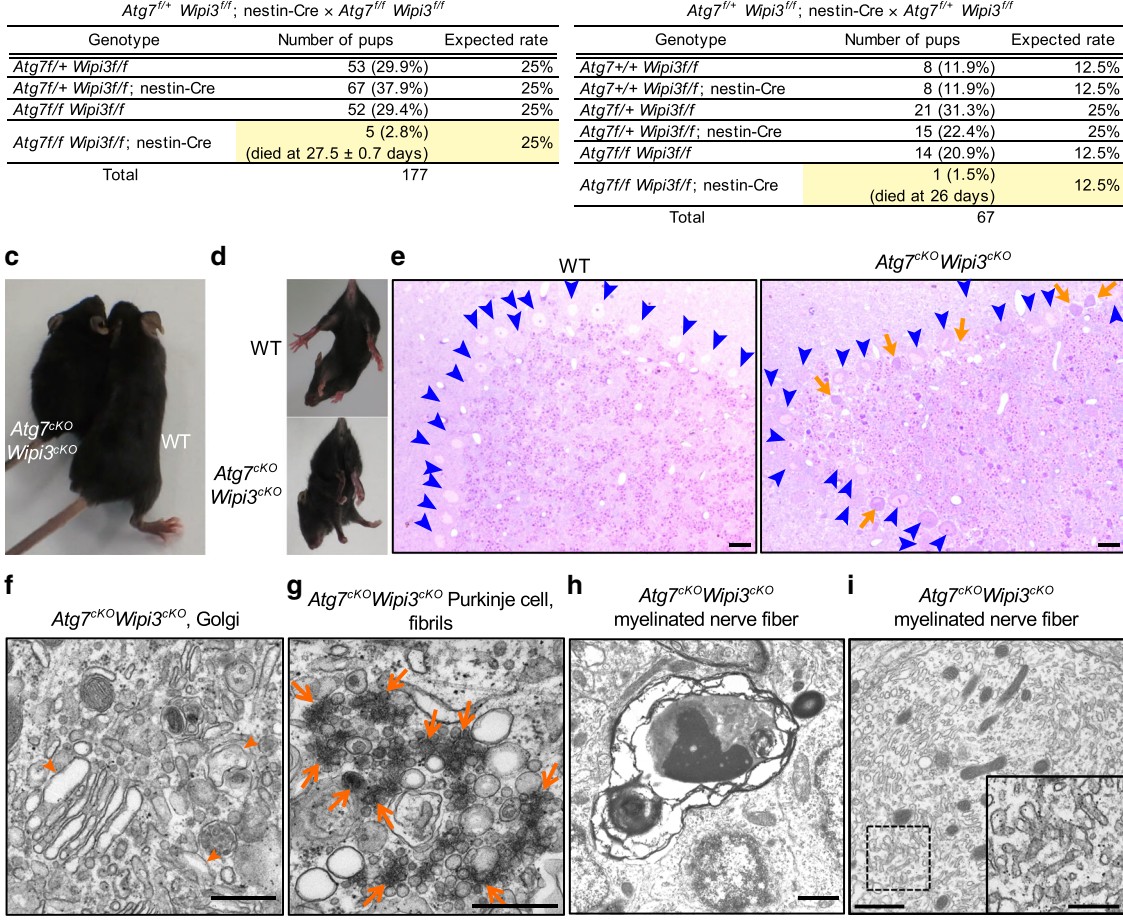

**Fig. 10 *Atg7cKO/Wipi3cKO* mice showed a more severe phenotype than single-knockout mice. a, b** Tables showing the rate of viable neonates of the different genotypes. "Expected rate" indicates the Mendelian ratio. Yellow rows indicate the population of *Atg7cKO/Wipi3cKO* mice. **c** Gross appearance of the *Atg7cKO/Wipi3cKO* mice at 4 weeks of age. **d** The limb-clasping reflex is observed in *Atg7cKO/Wipi3cKO* mice. **e** Histological analyses of *Atg7cKO/Wipi3cKO* mice. Semi-thin sections of the cerebellum were stained with toluidine blue. Bars = 20 μm. Blue arrowheads and orange arrows indicate live and dead Purkinje cells, respectively. **f–i** EM analysis of *Atg7cKO/Wipi3cKO* mouse brains. **f** Swollen rod-shaped Golgi membranes were evident in the Purkinje cells from *Atg7cKO/Wipi3cKO* mice (arrowheads). The morphology was similar to that of *Wipi3cKO* mouse brains (Fig. 7i). **g** The accumulation of fibrils was observed in Purkinje cells from *Atg7cKO/Wipi3cKO* mice (arrows), as observed in *Wipi3cKO* Purkinje cells (Fig. 7k). **h, i** Myelinated nerve fibers of *Atg7cKO/Wipi3cKO* cerebella were filled with debris (**h**) and smooth ER (**i**). Bars = 0.5 μm in (**f, g**) and 1 μm in (**h, i**). Magnified images of the dashed squares are shown in the insets. Bar = 0.5 μm.

and anti-DMT1 (Proteintech, #20507-1-AP). Enzymes used for recombinant DNA techniques were purchased from Takara Bio, TOYOBO, and New England Bio-Labs. All other reagents were obtained from Nacalai Tesque.

**Yeast strains and media**. The yeast strains used in this study are listed in Table S1. Gene deletions were introduced into the yeast by homologous recombination. Transformation into yeast was performed using the standard lithium acetate method. For AmphoB treatment, yeast cells were cultured in YPD medium to the stationary growth phase, and the cells were shifted to YPD medium containing AmphoB for 24 h with shaking at 30 °C.

**Electron microscopy**. Yeast and mammalian cells were placed between sapphire glasses (Niko Optical) or aluminum plates and frozen quickly by soaking in liquid nitrogen. Frozen samples were fractured in liquid nitrogen and fixed by the quick freeze-substitution method[47], which achieves good preservation of cellular architecture.

Mammalian cells and brains were fixed by a conventional fixation method (1.5% paraformaldehyde and 3% glutaraldehyde in 0.1 M phosphate buffer [pH 7.4], followed by an aqueous solution of 1% osmium tetroxide). Fixed samples were embedded in Epon 812, and thin sections (70–80 nm) were then cut and stained with uranyl acetate and lead citrate for observation under a Jeol-1010 electron microscope (Jeol) at 80 kV.

**GFP processing assay in yeast cells**. Yeast cells with GFP-pho8Δ60 were precipitated with trichloroacetic acid and washed with ice-cold acetone. Then, cells were lysed by vertexing with acid-washed glass beads and used for western blot analysis[9].

**Mice and embryonic fibroblasts**. To generate *Wipi3<sup>flox</sup>* mice, the ES cell line targeting the *wipi3* gene was purchased from the EUCOMM. The ES cell clones were microinjected into blastocysts to generate chimeras. The chimeric mice were screened and intercrossed with flippase transgenic mice. Then, *Wipi3<sup>flox</sup>* mice were crossed with C57BL/6 mice for at least five generations. The *Atg5<sup>KO</sup>* and *Atg7<sup>flox</sup>* mice were described in other studies[18,48]. The nestin-cre mice were purchased from the Jackson Laboratory. Mice were bred in a 12-h light/12-h dark cycle at ~23 °C and 40% relative humidity at the Laboratory for Recombinant Animals of Tokyo Medical and Dental University, Tokyo, Japan. This animal facility is operated according to the NIH guidelines. The Tokyo Medical and Dental University Ethics Committee for Animal Experiments approved all experiments in this study, and all experiments were performed according to their regulations. We used only male mice in these experiments.

From embryos at E13.5, we generated WT, *Atg5<sup>KO</sup>*, *Wipi3<sup>flox</sup>*, and *Atg7<sup>flox</sup>/Wipi3<sup>flox</sup>* MEFs and immortalized them with the SV40 T antigen. The *Wipi3<sup>KO</sup>* and *Atg7<sup>KO</sup>/Wipi3<sup>KO</sup>* MEFs were then generated from the *Wipi3<sup>flox</sup>* and *Atg7<sup>flox</sup>/Wipi3<sup>flox</sup>* MEFs by retroviral transfection of Cre recombinase.

The *Wipi3<sup>Cr</sup>*, *Atg5<sup>KO</sup>/Wipi3<sup>Cr</sup>* MEFs, *Atg5<sup>Cr</sup>/Ulk1<sup>KO</sup>/Ulk2<sup>KO</sup>* MEFs, *Wipi2<sup>Cr</sup>* MEFs, and *Atg5<sup>KO</sup>/Wipi2<sup>Cr</sup>* MEFs were generated from WT, *Atg5<sup>KO</sup>* MEFs, and *Ulk1<sup>KO</sup>/Ulk2<sup>KO</sup>* MEFs by the CRISPR/Cas9 system[49]. In brief, a 20-bp mouse Wipi3-targeting sequence (AGCGAAATAACATTTCCACA), Atg5-targeting sequence (AAGAGTCAGCTATTTGACGT), and Wipi2-targeting sequence (CAGAGCGGAGAGGCCGGCGC) was synthesized (Eurofins) and introduced into the px330 plasmid (Addgene). This plasmid together with a hygromycin resistance plasmid was transfected into MEFs, and single clones were picked after hygromycin selection (200 μg/mL). The deletion of *Wipi3, Atg5,* and *Wipi2* was confirmed by genomic sequencing. Details are shown in Supplementary Figs. 1, 4, 15 and 16. MEFs were cultured in Dulbecco's modified Eagle's medium with 10% fetal bovine serum as described[23].

**Plasmids**. The plasmids used in this study are listed in Table S2. For stable DNA transfection, retroviral expression plasmids were generated and introduced into MEFs using plat-E cells[9]. For transient DNA transfection, we introduced plasmids into MEFs with the Neon electroporation system (Invitrogen) according to the manufacturer's instructions. The primers used are listed in Supplementary Table3.

**Alternative autophagy assay**. The generation of autolysosomes was analyzed using mRFP-GFP and mRFP-GFP-Rab9 proteins. In brief, MEFs expressing these proteins were observed using confocal microscopy, and were fixed in 4% paraformaldehyde, permeabilized in 0.1% saponin, and immunostained with an anti-Lamp2 antibody. The cells were then mounted in 4',6-diamidino-2-phenylindole (DAPI) and observed by confocal microscopy.

Simple immunostaining using an anti-Lamp2 antibody was also useful to identify autolysosomes. This is because Lamp2 fluorescence is seen as tiny dots in nonautophagic cells, whereas it forms ring-like structures in autophagic cells, as shown previously[5,9]. In all these autophagy assays, we considered alternative autophagic cells as cells with more than one puncta (red puncta from mRFP-GFP, mCherry-puncta, and ring-like Lamp2 puncta with a bigger diameter than 1 μm).

For the mCherry-Rab9 cleavage assay, lysates from mCherry-Rab9-expressing MEFs were electrophoresed and western blot analysis was performed using an anti-RFP antibody.

**STED microscopy**. Cells on glass coverslips (Matsunami, No.1 S; 160- to 190-μm thick) were fixed in 0.75% paraformaldehyde and 1% glutaraldehyde in HBSS buffer. After washing twice with PBS, cells were incubated with 100 mM glycine in PBS for 30 min, followed by incubation with 10 mg/mL sodium borohydrate and 100 mM ammonium chloride in PBS (pH 8.0) for 40 min. Cells were permeabilized in 0.1% Triton X-100 in PBS, and stained with primary antibodies followed by Alexa fluor 488 or Alexa fluor 555 secondary antibodies (Thermo Fisher Scientific). Cells were mounted in mounting medium with ProLong Gold antifade reagent (Thermo Fisher Scientific). STED images were recorded with a gSTED (TCS SP8 STED 3X, Leica Microsystems). Fluorescent dyes were excited with 488 nm and 561 nm laser lines from white laser, depleted with 660 nm STED lazer (donut beam) and recorded with HyD detectors under the photon counting mode. Raw STED images were recorded with a 100 Å~oil immersion objective (Leica HCX PL APO 100 Å~/1.4, oil), and deconvolved by using a software (Huygens Professional, SVI).

**Duolink in situ proximity ligation assay**. Cells were fixed in 4% paraformaldehyde containing 8 mM EGTA for 10 min and then permeabilized using 50 μg mL⁻¹ digitonin for 5 min. Cells were then stained with the indicated primary antibodies overnight at 4 °C. After washing, the cells were assayed with Duolink in situ reagents according to the manufacturer's instructions, and mounted in Prolong Gold Antifade reagent with DAPI (Thermo Fisher Scientific) and a laser-scanning confocal microscope (LSM710, Zeiss). Data analysis was performed using Zen and Image J software.

**Immunofluorescence assay**. MEFs were transiently expressed 3×Flag-Wipi2, 3 × Flag-Wipi3 or its mutants together with mRFP-GFP, HA-LC3, HA-GS15 (a trans-Golgi marker), and mCherry-×FYVE proteins. After various treatments, their fluorescence was observed in living cells. Then, the MEFs were fixed, permeabilized, and immunostained with anti-Flag, anti-HA, and anti-Lamp2 antibodies. The cells were then mounted with DAPI, and observed by confocal microscopy.

**Subcellular fractionation**. Cells were washed with PBS twice and then incubated in 2 mL of ice-chilled lysis buffer (10 mM HEPES-KOH (pH 7.4), 12.5% sucrose, 1 mM EDTA, 1 mM DTT, 20 μg/mL PMSF, 5 μg/mL antipain, 1 μg/mL aprotinin, 0.5 μg/mL leupeptin, and 0.7 μg/mL pepstatin) for 10 min and then homogenized 20 times with a Dounce homogenizer. The homogenate was centrifuged twice at $500 \times g$ for 15 min to obtain the postnuclear supernatant. The postnuclear supernatant was layered on top of a 20–60% sucrose gradient (10 mM HEPES-KOH [pH 7.4], 1 mM MgCl₂) and ultracentrifuged at $200,000 \times g$ for 2.5 h at 4 °C. The 12 resulting fractions were collected from top to bottom.

**Electron microscopy of subcellular fractionation samples**. Subcellular fractionation samples were fixed in the 1.5-mL tube or on a sapphire glass (Niko optical). The fixative used was the same as that used for the cell fixation as described below in Electron microscopy. For immunolabeling of subcellular fractionated samples, fraction samples (80 μL) were placed on a sapphire glass (Niko optical) and primary antibodies were added (final dilution: 1/200), incubated for 4 h at 30 °C, followed by incubation with immunogold-labeled anti-rabbit IgG and anti-mouse IgG antibodies (final dilution: 1/100) for 1.5 h at 30 °C. After the solution was absorbed using filter paper, the sapphire glass was soaked in the fixative (the same as described above).

**VSVG trafficking**. MEFs were transfected with a vector expressing VSVG–GFP and plated, and then treated with or without etoposide (10 μM) at the restrictive temperature for VSVG–GFP (40 °C) for 12 h. Cells were then shifted to the permissive temperature (32 °C), and fixed at the indicated times. Golgi and lysosomes were counterstained with an anti-GS28 and anti-Lamp2 antibodies, respectively.

For FACS analysis, cells were transfected with VSVG–GFP, and treated with or without etoposide (10 μM) at the permissive temperature (32 °C) for 60 min. Cells were then stained with an anti-VSVG antibody for 30 min at 4 °C without membrane permeabilization. After washing, the cells were stained with an Alexa Fluor 633-conjugated secondary antibody for 30 min. After washing, cells were analyzed by flow cytometry. Data analysis was performed using BD FACSDiva and FlowJo software.

**Rotarod analysis of mice**. Male mice were tested on an accelerating rotarod apparatus (Ugo Basile) set to accelerate from 4 to 40 rpm over a period of 300 s[50].

**Histological analysis**. Mice were fixed by cardiac perfusion with 0.1 M phosphate buffer containing 4% paraformaldehyde and 4% sucrose for light microscopy, or with 0.1 M phosphate buffer containing 1.5% paraformaldehyde and 3% glutaraldehyde for EM. Brain tissues were excised and processed for morphological analysis. For light microscopic analysis, 10-μm cryosections were cut and immunolabelled with anti-GFAP (Sigma) and anti-calbindin (Sigma) antibodies.

**Iron staining**. To detect iron accumulation, an iron staining kit (ScyTek) was used according to the manufacturer's protocol. In brief, the potassium ferrocyanide solution was mixed with an equal volume of hydrochloric acid solution. Mouse brain cryosections were incubated with mixed solution, followed by nuclear staining. Sections were examined under an Olympus BS2 microscope using DP controller (Olympus).

**Lentivirus production**. Lentivirus vectors (pRRL-cPPT-CMV-X2-PRE-SIN) expressing 3×Flag-Wipi3 (Lenti-Wipi3) or 3×Flag-Dram1 (Lenti-Dram1) were constructed, and the vectors were transfected into HEK293 cells together with a packaging plasmid (pCAG-HIVgp) and envelope plasmid (pCMV-VSV-G-RSV-Rev) using polyethylenimine. Cells were harvested for 48 h after the transfection, and the lentivirus-containing supernatants were passed through a 0.22-μm PVDF filter. Then, the supernatants were concentrated 1000-fold by ultracentrifugation at $6000 \times g$ for 16 h. The final purified lentiviruses were aliquoted and stored at −80 °C. The titers of the purified lentiviruses were calculated using Lenti-X p24

rapid titer kit (Clontech), and the titers were as follows: Lenti-Vector, $4.9 \times 10^9$; Lenti-Wipi3, $3.6 \times 10^9$; and Lenti-Dram1, $1.5 \times 10^9$ IU/mL.

**Stereotaxic lentivirus injections**. Five-week old male *Wipi3cKO* mice were deeply anesthetized with a mixture of medetomidine hydrochloride (0.75 mg/kg), midazolam (4 mg/kg), and butorphanol tartrate (5 mg/kg) and placed on a stereotaxic apparatus (Narishige). The skull was exposed, and mice were bilaterally injected with 2.0 μL ($1.5 \times 10^9$ IU/mL) of either Lenti-Vector, Lenti-Dram1, or Lenti-Wipi3 into the cerebellum over 20 min. The coordinates relative to bregma were, anteroposterior: −6.30 mm, −7.80 mm; lateral, ±1.50 mm; dorsoventral, −1.50 mm (minus indicates a position posterior to bregma).

**Correlative light and electron microscopy (CLEM)**. For the merging of confocal fluorescence microscopy photos and transmission electron microscopy (TEM) photos, cells were cultured on coverslips with grids (Matsunami) and fixed with 0.75% paraformaldehyde/1.5% glutaraldehyde in PBS for 15 min at room temperature. Subsequently, the samples were visualized by confocal microscopy; and then fixed using 1% $OsO_4$ at 4 °C for 15 min and examined by TEM.

**Cell viability assay**. Cells were stained with propidium iodide (PI) and cell viability was detected by flow cytometry (BD; FACS Canto II, Supplementary Fig. 29). Data analysis was performed using BD FACSDiva and FlowJo software.

**Protein expression and purification**. GST-fusion proteins were purified from *E. coli* JM109 cells containing the respective plasmids. The cells were cultured at 16 °C and induced with 0.2 mM IPTG. After 15 hr, the cells were collected and washed with wash buffer (20 mM HEPES-KOH [pH 8.0], 160 mM potassium acetate). The total cell lysates were prepared by sonication in lysis buffer (20 mM HEPES-KOH [pH 8.0], 160 mM potassium acetate, 1 % Triton X-100, 1 mM EDTA). After removal of insoluble materials by centrifugation, the supernatant was incubated with glutathione-sepharose beads (GE Healthcare) at 4 °C for 1 h. The beads were washed by wash buffer, and the GST-fusion proteins were eluted with elution buffer (100 mM Tris-HCl [pH 8.0], 20 mM reduced glutathione).

**Lipid protein overlay assay**. Two-fold dilutions of PI(3)P solubilized in chloroform were spotted onto Hybond-C extra membrane (Amersham Life Sciences) as indicated and air dry for 1 h at room temperature. The membranes were blocked with blocking buffer [3% (w/v) BSA, 1 mM $MgCl_2$ in TBS-T (50 mM Tris-HCl [pH 7.5], 150 mM NaCl, 0.1% Tween-20)] and incubated with 0.5 μg/mL indicated GST-fusion proteins at 4 °C overnight. Membranes were washed with wash buffer (1 mM $MgCl_2$ in TBS-T), and the bound proteins were detected by immunoblotting using anti-GST (final dilution: 1/500).

**Statistical analysis**. Results are expressed as the mean ± standard deviation (SD) or standard error of the mean (SEM). Statistical analyses were performed using Prism8 (GraphPad) software. Comparisons of two data sets were performed using unpaired two-tailed Student *t*-tests. All other comparisons of multiple datasets were performed using one-way ANOVA followed by the Tukey post hoc test. A *p*-value < 0.01 or 0.05 was considered to indicate a statistically significant difference between two groups.

**Statistics and reproducibility**. Repeated independent experiments per each panel with similar results are shown below. n = 1 (Fig. 8a–e, Supplementary Figs. 3a, 4b, 9a, c, 16a, c, d, 24a–c, 25c, 26b, c, 27a, b, 28a, b). n = 2 (Fig. 6c, f, g, Supplementary Figs. 2c, 4c, 12, 14, 15c, 16e–g, 18a–c, 19a, b, 20a–d, 21, 22, 26a). n = 3 (Fig. 6b, d, e, h, i, Supplementary Figs. 1b, 2a, b, 3b, 9b, d, 15b).

**Reporting summary**. Further information on research design is available in the Nature Research Reporting Summary linked to this article.

## Data availability

All data that supporting the findings of this study are available from the corresponding author upon request. The source data underlying Figs. 1–10 and Supplementary Figs. 1–29 are provided as a Source Data file. Source data are provided with this paper.

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

## Acknowledgements
We are grateful to Dr. A. Nakano (The University of Tokyo) for helpful discussions. We are also grateful to Drs. N. Mizushima (The University of Tokyo), S. Tooze (The FRANCIS CRICK Inc.), and T. Kitamura (The University of Tokyo) for kindly providing us with the *Atg5KO* mice, *Ulk1/Ulk2DKO* MEFs, and plat-E cells, respectively. We are also grateful to Drs. M. Komatsu (Niigata University) and K. Tanaka (Tokyo Metropolitan Institute of Medical Science) for kindly providing *Atg7flox* mice. We also thank NPO Biotechnology Research and Development for technical assistance. This study was supported in part by a Grant-in-Aid for Scientific Research (A) (17H01533, 20H00467), Grant-in-Aid for Scientific Research (C) (19K07382, 20K07353), Grant-in-Aid for Young Scientists (B) (17K15448), Grant-in-Aid for challenging Exploratory Research (16K15230), Grant-in-Aid for Scientific Research on Innovative Areas (17H06413, 17H06414, 20H05314) from the MEXT of Japan. This study was also supported in part by the Project for Psychiatric and Neurological Disorders (JP20dm0107136), by the Project for Cancer Research and Therapeutic Evolution (P-CREATE) (JP20cm0106109), and by the Practical Research Project for Rare / Intractable Diseases (JP20ek0109407) from the Japan Agency for Medical Research and development (AMED). This study was also supported by the Joint Usage/Research Program of Medical Research Institute, Tokyo Medical and Dental University.

## Author contributions
H.Y. designed the research and performed the biochemical and cell biological analyses, S.A. designed the research and performed the EM analyses, S.H., S.T., N.F., and H.T.S. performed the experiments using mice, K.S., K.M., and N.M. performed the gene transfection experiments in mice, K.K. performed the STED analysis, and S.S. designed the research and wrote the paper.

## Competing interests
The authors declare no competing interests.
