## [Peer Review File · Nature Communications]

Reviewers' comments:

Reviewer #1 (Remarks to the Author):

The manuscript by Yamaguchi et al. is a monumental effort that contains a great deal of advances in many aspects of our understanding of autophagy and is well worthy of consideration for publication.

It is uneasy to ask for further clarifications, given the vastity of the paper, but precisely because of the wide spectrum covered, there are gaps left that I believe should be filled to solidify the paper, which could well be two papers, in fact.

First, there is a discrepancy in the approach followed in the in vitro cellular part and the in vivo part in mouse. All experiments in the first part are essentially made in double knock out cells (Atg5 or Atg7 and Wipi3), while the analysis in mouse is purely on Wipi3 knock out. As the authors observe that double knock-out mice are hardly produced, this suggest that the cellular phenotypes observed could be arising from this double deficiency. The rescue experiments may not entirely escape this criticism, as perhaps overexpression of Wipi3 could mask or stimulate additional pathways. Given that the authors have the possibility of using Wipi3 knock-out MEFs they should provide a set of experiments to validate that the defects observed in the double ko cells, are indeed present in the Wipi3 ko only, and that would make a much easier and straightforward comparison to the in vivo mouse part. A complementary point to this is that it would be valuable to gather any data on the cellular pathology of the double ko mice, even if that would be limited to a few individuals, to evaluate how much of the in vitro cellular phenotypes are recapitulated in vivo.

Second point, the obvious key question in this alternative autophagy is: what is it for? What are the fibrils that are specifically degraded? Given that alternative autophagy is induced by a DNA damaging drug, could those fibrils be DNA and chromatin? And if so, could this uncover a process of nucleophagy, separate from the LC3-dependent nucleophagy that has been described? I believe the authors are well placed to at least test this hypothesis and try and nail down what the key target of this alternative autophagy is.

In addition to these main points that may require additional experiments, there are a few points that should be discussed in the manuscript

- As the authors point out, yeast Hsv2 is orthologue to both Wipi3 and Wipi4, however the authors do not address at all Wipi4. Why? Could they rule out that it also may have a role in alternative autophagy?
- Re the role of Wipi3, given the abnormal shape of the Golgi caused by its deficiency, can the authors rule out that what Wipi3 is really needed for, is for the Golgi to maintain its proper shape? If this shape is lost, then the Golgi membranes are unable to function properly and among other things they may be unable to perform alternative autophagy. If this hypothesis is correct, then also other Golgi functions may be disrupted by Wipi3 deficiency. Can the authors rule this out?
- With respect to the Kikume technique, in Supplementary Figure 3c, many, but not all, punctae overlap with LC3. Are those LC3 negative punctae alternative autophagy or autolysosomes? Would they be positive for Lamp1?
- Statistical tests should be detailed in the figure legends.

- For the mouse data, I could not find any indication of whether both males and females were used or only one sex and whether any difference was observed. This is an important information to report, to be able to evaluate any gender bias in the analysis.
- In vivo, GOMED autophagy was reported in the cerebellum of mouse models for a polyglutamine disease, in parallel to nucleophagy, perhaps another suggestion of a link between the two processes?

Reviewer #2 (Remarks to the Author):

The manuscript by Yamaguchi and colleagues investigates the role of Wipi3 in alternative autophagy, a recently identified pathway that is independent of Atg5 and Atg7. The authors report that Wipi3 binds to Golgi membranes and is required to form isolation membranes in cells treated with the drug etoposide. The importance of Wipi3 is investigated through the analysis of Wipi3-deficient mice, which show behavioral defects and loss of cerebellar neurons.

There is no question that this manuscript describes interesting findings that are of potential significance and will be of interest to the field. That being said, this is a surprisingly difficult manuscript to read or appreciate. The manuscript includes extensive experimental data in both the main text and the supplemental figures. These figures are poorly organized and overly complex. More importantly, the work has a broad reach but sometimes insufficient depth. For example, the work runs from initial screens in yeast through molecular and cellular assays to generation of a Wipi3-deficient mouse and a brief description of behavioral and pathological assays. With these many results reported in a single manuscript, it seems ridiculous to ask for more. However, many of the points made lack depth. This lack of rigor on specific points leads one to worry that some of the conclusions from the individual experiments are overstated, with multiple explanations possible from the results, not just those indicated. The sheer number of assays make the manuscript hard to understand and follow. Due to the space constraints on the text and the volume of data, little is introduced appropriately and transitions between experiments explaining the logic are sorely lacking.

My best advice to the authors would be to consider breaking this unwieldy manuscript into two papers. One could define the role of Wipi3 at the cellular level in the alternative autophagy pathway, and one could define the role of Wipi3 in vivo using the mouse model they have developed. This would allow them to more fully establish the many points they wish to make in a more convincing manner.

If they do not wish to divide this work, then at a minimum it should be restructured so that each figure includes the relevant data to make a particular point. Within the text, more care should be taken to introduce each assay, and to thoroughly interpret the results and possible caveats of these interpretations.

For example, the authors report that Wipi3 is recruited to the Golgi via binding to PI3P. But there are many closely related PI3P binding proteins expressed in the cell, including Wipi2. Why do the

authors think that only Wipi3 is recruited and why can only Wipi3 function in this alternative autophagy pathway?

Additional specific points:

1. Despite the presence of “neuron” in the title of the paper, very few experiments are actually in neurons.
2. There is at least one other Wipi3 antibody that the authors did not mention in the text – from Santa Cruz. Have the authors tried this reagent or considered making their own antibody?
3. For Figure 1e, there are no controls shown (wild type cells, untreated).
4. For Fig 2J-K, why do the Rab9 channels look so different between NT and Wipi3Cr with etoposide?
5. What is the difference between Fig 3D and Fig 3H? If the y-axis graph label in Fig 3H is correct, the graph is very confusing, as there are many puncta in both conditions. If the graph is truly of total Wipi3 puncta, then Fig 3C and Fig S11 are not representative images, as both show many Wipi3 puncta in the NT condition. If the y-axis label in Fig 3H is wrong and should read “Cells with Golgi-localized Wipi3 puncta,” I don’t understand how Fig 3C and Fig 3H are different.
6. For the mRFP-GFP and Kik fluorescent micrographs, nearly all of them are overexposed, making it virtually impossible to determine anything from them. In addition, colocalization between yellow and cyan is impossible to see. Cyan should be changed to a different color.
7. More broadly, the Kikume assay seems of limited utility – it is simply a cytosol exclusion assay, and thus not actually specific for the autophagy compartment they are investigating.
8. In Fig. S3A-B, the micrographs are not of the same cell in each panel, which does not inspire confidence in the new alternative autophagy assay.
9. In Fig S9B, the n=4. This should be much higher for MEFs. For Fig S9C, why isn’t Wipi3Cr tested? Same question for Fig S9.
10. In line 190, “diffusely” should be changed to “evenly” – the signal is definitely punctate, not diffuse.
11. In line 195-196, “some Flag-Wipi3 appeared as cytoplasmic puncta together with the trans-Golgi” doesn’t make sense – is Wipi3 cytoplasmic, punctate, or in the trans-Golgi? I don’t think it can be all three at once. The same problem can be found in the figure legend. The Wipi3 puncta just look like background IF staining more than actual, biologically relevant puncta.
12. In line 196, in conjunction with Fig 3I, puncta are referred to as “Wipi3/trans-Golgi puncta,” but there is no Golgi marker in those micrographs. “trans-Golgi” should be removed.
13. For Fig S15F&H, why are the p62 results opposite while the LC3-II results are congruent?
14. The manuscript could not be fully assessed because data panels were missing for Fig S9D&E, Fig S10, and Fig S16C.
15. In Fig S22D, how do we know that the structures are mitochondria? They don’t look like mitochondria to me.
16. What are we supposed to get from Fig S23? The two micrographs are very different magnifications and the “structures” that the arrows point to look more like EM artifacts than real, biologically relevant formations.
17. Fig S26B-C should be moved to Fig 6.

In sum, this work is potentially of interest to the field, but the current manuscript is dense and difficult to read. The work is overly broad and insufficiently deep on some key points. And there are a range of issues, such as those detailed above, that raise concern about the ability of the authors to

clearly present each experiment and judiciously interpret each result in the forest of assays and experimental results reported here.

Reviewer #3 (Remarks to the Author):

In this manuscript, Yamaguchi et al address the role of WIPI3 in alternative autophagy both in intact cells and in vivo, after generating a neuron-specific conditional KO mouse. The authors begin their studies by focusing on yeast cells, where they have identified an experimental paradigm of noncanonical autophagy by knocking out Atg5 and inducing the alternative pathway with amphotericin B1. They identified a number of genes operating in the alternative autophagy pathway, including Hsv2, which is the ortholog of WIPI3 or 4. They decide to pursue WIPI3 in mammalian cells, because it is the PROPPIN isoform that appears to be the most relevant for alternative autophagy. The authors make a reasonable case for an involvement of WIPI3 in alternative autophagy, by using an Atg5 null background in combination with etoposide treatment and showing a requirement for this protein in a variety of morphological phenotypes. They finally generate a neuron-specific cKO mouse for WIPI3 and find a variety of phenotypes in the CNS, including loss of Purkinje neurons, astro- and microgliosis, increased presence of fibrils and swollen Golgi in Purkinje cells. A subset of phenotypes can be rescued by overexpressing with lentivirus the protein DRAM1, which is a Golgi protein also involved in alternative autophagy and can rescue WIPI3 loss-of-function phenotypes in cell culture.

Overall, this is an interesting study that documents a number of unexpected phenotypes in the WIPI3 KO cells in vitro and neurons in vivo. The morphological analyses presented in the study are spectacular, although there isn't a systematic morphometric analysis of key findings, making it difficult to assess how common or prevalent the phenotypes are. However, while the authors can clearly show a role for WIPI3 in alternative autophagy in vitro using artificial conditions, such as lack of key components of the classical autophagy (e.g. ATG5 or 7) combined with etoposide treatment, there is no evidence that specific defects in alternative autophagy are what accounts for the phenotypes in neurons in vivo. Both WIPI3 and DRAM1 are Golgi-associated proteins and lack of WIPI3 causes a Golgi phenotype in Purkinje cells which can be rescued with DRAM1. These findings do not prove that these proteins have any connections with alternative autophagy. In fact, it may simply show that messing up with the Golgi is deleterious for neurons. The only piece of evidence suggestive of a role of WIPI3 in some form of autophagy is the accumulation of fibrils in the KO Purkinje cells. However, there is no information on what those fibrillar structures may be and thus no direct way to test whether these structures arise from defects in alternative autophagy in vitro or in vivo. In the absence of this kind of evidence, the conclusions of this manuscript are largely overstated and the paper is more suitable for a specialized journal, after toning down the interpretations of the in vivo work. Unfortunately, the physiological role of alternative autophagy is not clarified by this study.

Minor comments:

- Lines 24-26: please be more specific and state what is different about the morphology, rather than

stating that they are different.

- Lines 39-40: The statement “Recently, this machinery was reported to work slowly even in the absence of Atg5” seems odd, as the autophagy machinery is precisely expected to work slowly in the absence of Atg5.

- Line 68: “is blocked” rather than “are blocked”.

Point-by-point responses to the Reviewers' comments

Responses to the comments by the Editor

Notably, while all reviewers find the manuscript to be of potential interest, they all share concerns insufficiently developed considerations including but not limited to the role of other potential phosphoinositide binding proteins (please see comments from Reviewers #1 and #2), unclear effects on alternative autophagy, and unclear Wipi3 localization and function (all Reviewers). While we do not necessarily require splitting into two manuscripts (as otherwise suggested by Reviewers #1 and #2), we would suggest focusing and strengthening current claims (especially the potential proposed molecular/cellular mechanisms) as detailed by the Reviewers. Please note we allow for 5000 words for Introduction, Results, and Discussion (with some flexibility); as well as up to 10 main figures; hopefully this will allow space to further address Reviewer concerns.

Response:

We appreciate these helpful comments. We have addressed all of the reviewers' comments in full. Particularly, in accordance with the comments from the editor and the reviewers, we analyzed the role of other potential phosphoinositide-binding proteins. Because Wipi2 is a protein containing a phosphoinositide-binding domain and is a homologue of Wipi3, we analyzed its role in alternative autophagy using a STED microscope. Interestingly, although Wipi2 contains a phosphoinositide-binding domain and its overall structure is the same as Wipi3, it did not translocate to the Golgi upon etoposide treatment (Suppl. Fig. 15a). Therefore, we considered that the phosphoinositide-binding domain is essential, but not sufficient, for the translocation of Wipi3 to the Golgi. Therefore, another crucial signal sequence for Golgi trafficking is thought to exist in Wipi3, but not in Wipi2.

Regarding Wipi3 localization, we have already shown its translocation from the cytosol to the Golgi. In the revised manuscript, we added data showing the clear localization of Wipi3 with autolysosomes (Figure 3g and 6g), which support the important role of Wipi3 in alternative autophagy.

We have also demonstrated the role of Wipi3-dependent alternative autophagy.

Because we previously suggested the possibility that the molecules delivered from the Golgi to the plasma membrane (PM) are preferentially degraded by alternative autophagy upon Golgi stress, we analyzed the involvement of Wipi3 in this mechanism. As expected, Wipi3-dependent alternative autophagy was found to play a role in the elimination of excess Golgi cargos (Figure 5 of the revised manuscript).

The most important advance in this revision was elucidation of the possible mechanism of neurodegeneration in neuron-specific Wipi3-deficient mice, which was asked by the reviewers. Through our extensive search, we identified iron deposition and ceruloplasmin accumulation. Ceruloplasmin is a molecule functioning in iron metabolism. Importantly, ceruloplasmin is delivered from the Golgi to the PM, and which is a cargo of alternative autophagy. Therefore, it is expected that the defect in alternative autophagy resulted in an abnormal increase in ceruloplasmin in the cytosol, which in turn induced abnormal iron deposition, resulting in a neurodegenerative disorder (Figure 8). This finding is important for understanding the pathophysiological function of Wipi3.

We additionally improved various data in the manuscript to strengthen our conclusion, and restructured each figure to clearly demonstrate a particular point. Within the text, we carefully described the introduction of each assay, as well as proper interpretations to all of the results.

Responses to the comments by Reviewer #1

General Comment: The manuscript by Yamaguchi et al. is a monumental effort that contains a great deal of advances in many aspects of our understanding of autophagy and is well worthy of consideration for publication. It is uneasy to ask for further clarifications, given the vastity of the paper, but precisely because of the wide spectrum covered, there are gaps left that I believe should be filled to solidify the paper, which could well be two papers, in fact.

Response:

We greatly appreciate these constructive comments. In accordance with the suggestion, we performed various experiments to fill the gaps within our previous manuscript, and added them to the revised manuscript. Although we did consider splitting this manuscript into two, we would like to publish these data as one manuscript, according to the editor's suggestion. Instead, we rewrote the manuscript in order to make it easier to read.

Comment #1: First, there is a discrepancy in the approach followed in the in vitro cellular part and the in vivo part in mouse. All experiments in the first part are essentially made in double knock out cells (Atg5 or Atg7 and Wipi3), while the analysis in mouse is purely on Wipi3 knock out. As the authors observe that double knock-out mice are hardly produced, this suggest that the cellular phenotypes observed could be arising from this double deficiency. The rescue experiments may not entirely escape this criticism, as perhaps overexpression of Wipi3 could mask or stimulate additional pathways. Given that the authors have the possibility of using Wipi3 knock-out MEFs they should provide a set of experiments to validate that the defects observed in the double ko cells, are indeed present in the Wipi3 ko only, and that would make a much easier and straightforward comparison to the in vivo mouse part. A complementary point to this is that it would be valuable to gather any data on the cellular pathology of the double ko mice, even if that would be limited to a few individuals, to evaluate how much of the in vitro cellular phenotypes are recapitulated in vivo.

Response #1:

We believe this is a very important point. Therefore, we showed data from Wipi3-single KO MEFs and brains, and Wipi3/Atg5 (or Atg7)-double KO MEFs and brains. In all the cells, we found common abnormalities, i.e., swollen and rod-shaped Golgi membranes. These results suggested that Wipi3 targets the Golgi membranes.

We also described the morphological similarities between Wipi3-single KO MEFs and brains, as well as Wipi3/Atg5-double KO MEFs and Wipi3/Atg7-double KO brains. We showed all of these images in the Figures (Fig. 2b, Fig. 3b, Fig. 6h, Fig. 7, and Fig. 10), and summarized the results in the second section of the Discussion section of the revised manuscript.

Comment #2: Second point, the obvious key question in this alternative autophagy is: what is it for? What are the fibrils that are specifically degraded? Given that alternative autophagy is induced by a DNA damaging drug, could those fibrils be DNA and chromatin? And if so, could this uncover a process of nucleophagy, separate from the LC3-dependent nucleophagy that has been described? I believe the authors are well placed to at least test this hypothesis and try and nail down what the key target of this alternative autophagy is.

Response #2:

We believe this is also a very important point. To understand the role of Wipi3 and alternative autophagy, it is useful to identify the cargos that are accumulated in Wipi3-KO cells and mice. Reviewer #3 also raised the same point.

The reviewer kindly proposed the possible involvement of nucleophagy and recommended us to analyze DNA and nuclear factors. Thus, we investigated this possibility by staining DNA and nuclear proteins in MEFs and mice. However, we did not observe any apparent differences between Wipi3-expressing cells and Wipi3-deficient cells, at least in our experimental conditions. Therefore, these data are shown in this letter (Attached Figure 1) but not in the revised manuscript, owing to the space limitation of the manuscript.

We also focused on molecules delivered from the Golgi to the plasma membrane (PM), because we previously showed that these molecules are preferentially degraded by

alternative autophagy. As expected, VSVG-GFP, a molecule delivered from the Golgi to the PM, was degraded by Wipi3-dependent alternative autophagy upon etoposide treatment, indicating that the elimination of excess Golgi cargos is an important function of Wipi3-dependent alternative autophagy. These data are included in Figure 5 of the revised manuscript.

In *Wipi3^{CKO}* mouse brains, we also found iron deposition and ceruloplasmin accumulation. Ceruloplasmin is a molecule functioning in iron metabolism, and importantly, it was degraded by Wipi3-dependent alternative autophagy. From these observations, a basal level of alternative autophagy is expected to degrade excess ceruloplasmin to prevent iron deposition in the brain, and its failure is thought to induce abnormal iron deposition and a neurodegenerative disorder. These data are included in Figure 8 of the revised manuscript.

Comment #3: In addition to these main points that may require additional experiments, there are a few points that should be discussed in the manuscript

- As the authors point out, yeast Hsv2 is orthologue to both Wipi3 and Wipi4, however the authors do not address at all Wipi4. Why? Could they rule out that it also may have a role in alternative autophagy?

Response #3:

We also generated *wipi4*-deficient mice and MEFs. Wipi4 also functions in alternative autophagy *in vitro*, but the effect is weaker than that of Wipi3. Consistently, *wipi4*-deficient mice developed a neurodegenerative disorder, but the phenotypes were very mild, which is also reported by others (Zhao et al., Autophagy 2015). Because of the weak function of Wipi4 and the presence of a large amount of data regarding Wipi3, we focused on Wipi3 in this manuscript. Wipi4 was described only briefly in the revised manuscript, as follows (page 24, lines 6-11):

“SENDA is a human neurodegenerative disorder characterized by iron deposition, and is caused by a *wipi4* gene mutations. Our preliminary studies suggested that Wipi4 also functions in alternative autophagy, but to a smaller extent than Wipi3. Consistently, although *wipi4*-deficient mice developed a neurodegenerative disorder, the phenotypes

were much milder than those of *wipi3*-deficient mice, and they did not show substantial iron accumulation.”

Comment #4: - Re the role of Wipi3, given the abnormal shape of the Golgi caused by its deficiency, can the authors rule out that what Wipi3 is really needed for, is for the Golgi to maintain its proper shape? If this shape is lost, then the Golgi membranes are unable to function properly and among other things they may be unable to perform alternative autophagy. If this hypothesis is correct, then also other Golgi functions may be disrupted by Wipi3 deficiency. Can the authors rule this out?

Response #4:

The effect of Wipi3 on the Golgi is minimal in untreated cells, because Wipi3 is mostly distributed in the cytosol. Upon etoposide treatment, Wipi3 translocates to and manipulates the Golgi membrane. We observed a series of Golgi membrane alterations, including ministacked Golgi formation, elongation of *trans*-Golgi membranes, and autophagosome generation in Wipi3-expressing cells. In contrast, elongated *trans*-Golgi membranes were not observed in Wipi3-deficient cells, indicating that Wipi3 is crucial for the elongation of *trans*-Golgi membranes. We think the possible biochemical mechanism of Wipi3 is the elongation of membranes via its lipid-transfer activity. However, we do not formally deny the possibility that the nature of Golgi membranes is altered by the recruitment of Wipi3. Therefore, we described this notion in the fourth section of Discussion, as follows: (page 22, lines 20-23):

“Analogously, Wipi3 also interacts with *trans*-Golgi membranes using its PI3P interaction motif upon etoposide treatment, which may function in the elongation of isolation membranes from Golgi membranes for alternative autophagy. Alternatively, Wipi3 may alter the nature of Golgi membranes for their elongation. ”

The reviewer also suggested the possible disturbance of Golgi function by a lack of Wipi3, and hence we analyzed Golgi trafficking ability using VSVG-GFP. The delivery efficiency of VSVG-GFP was only slightly reduced in Wipi3-deficient MEFs (please

compare image 3 and 9 in Fig. 5b, and image 1 and 4 in Fig. 5d), suggesting that Wipi3 may weakly affected Golgi trafficking function.

Comment #5: - With respect to the Kikume technique, in Supplementary Figure 3c, many, but not all, punctae overlap with LC3. Are those LC3 negative punctae alternative autophagy or autolysosomes? Would they be positive for Lamp1?

Response #5:

Because starvation mainly induces canonical autophagy, most of the LC3-negative Kikume puncta in Suppl. Fig. 3c of the original manuscript are thought to be autolysosomes. However, owing to the immature validation of the Kikume assay (as pointed out by another reviewer) and space limitation, we removed all the Kikume data from the revised manuscript.

Comment #6: - Statistical tests should be detailed in the figure legends.

Response #6:

In accordance with the reviewer's comment, we consulted a statistics specialist, and described the statistical tests in the figure legends of the revised manuscript.

Comment #7: - For the mouse data, I could not find any indication of whether both males and females were used or only one sex and whether any difference was observed. This is an important information to report, to be able to evaluate any gender bias in the analysis.

Response #7:

We used only male mice, and we have added this information to the Materials and Methods section of the revised manuscript.

Comment #8: - In vivo, GOMED autophagy was reported in the cerebellum of mouse models for a polyglutamine disease, in parallel to nucleophagy, perhaps

another suggestion of a link between the two processes?

Response #8:

We thank you for this important suggestion. We missed referring to the important paper **“Stall in canonical autophagy-lysosome pathways prompts nucleophagy-based nuclear breakdown in neurodegeneration”** by Baron et al., showing the involvement of alternative autophagy in the polyQ diseases. We have referred to this paper in the revised manuscript.

Responses to the comments by Reviewer #2

General Comment: The manuscript by Yamaguchi and colleagues investigates the role of Wipi3 in alternative autophagy, a recently identified pathway that is independent of Atg5 and Atg7. The authors report that Wipi3 binds to Golgi membranes and is required to form isolation membranes in cells treated with the drug etoposide. The importance of Wipi3 is investigated through the analysis of Wipi3-deficient mice, which show behavioral defects and loss of cerebellar neurons. There is no question that this manuscript describes interesting findings that are of potential significance and will be of interest to the field. That being said, this is a surprisingly difficult manuscript to read or appreciate. The manuscript includes extensive experimental data in both the main text and the supplemental figures. These figures are poorly organized and overly complex. More importantly, the work has a broad reach but sometimes insufficient depth. For example, the work runs from initial screens in yeast through molecular and cellular assays to generation of a Wipi3-deficient mouse and a brief description of behavioral and pathological assays. With these many results reported in a single manuscript, it seems ridiculous to ask for more. However, many of the points made lack depth. This lack of rigor on specific points leads one to worry that some of the conclusions from the individual experiments are overstated, with multiple explanations possible from the results, not just those indicated. The sheer number of assays make the manuscript hard to understand and follow. Due to the space constraints on the text and the volume of data, little is introduced appropriately and transitions between experiments explaining the logic are sorely lacking.

My best advice to the authors would be to consider breaking this unwieldy manuscript into two papers. One could define the role of Wipi3 at the cellular level in the alternative autophagy pathway, and one could define the role of Wipi3 in vivo using the mouse model they have developed. This would allow them to more fully establish the many points they wish to make in a more convincing manner.

Response:

We greatly appreciate these constructive comments. As proposed by the Reviewer, we performed various experiments to fill the gaps within our previous manuscript,

reorganized the findings, and extensively rewrote the manuscript so that it is easier to read.

We also considered splitting this manuscript, as we received the same comment from Reviewer #1. However, we would like to publish these data as one manuscript, according to the editor's suggestion.

Comment #1: If they do not wish to divide this work, then at a minimum it should be restructured so that each figure includes the relevant data to make a particular point. Within the text, more care should be taken to introduce each assay, and to thoroughly interpret the results and possible caveats of these interpretations.

Response #1:

In accordance with these comments, we restructured each figure to clear a particular point. Within the text, we carefully explained each assay, and provided proper interpretations of the results.

Comment #2: For example, the authors report that Wipi3 is recruited to the Golgi via binding to PI3P. But there are many closely related PI3P binding proteins expressed in the cell, including Wipi2. Why do the authors think that only Wipi3 is recruited and why can only Wipi3 function in this alternative autophagy pathway?

Response #2:

Because Wipi2 is a protein containing a phosphoinositide-binding domain and is a homologue of Wipi3, we compared its behavior with using STED microscopy. Interestingly, although Wipi2 contains a phosphoinositide-binding domain and its overall structure is the same as Wipi3, Wipi2 did not translocate to the Golgi upon etoposide treatment (Suppl. Fig. 15a). Therefore, we considered that the phosphoinositide-binding domain is essential, but not sufficient, for the translocation of Wipi3 to the Golgi. Another crucial signal sequence for Golgi trafficking is thought to exist in Wipi3, but not in Wipi2. Many other PI3P-interacting molecules are not translocated to the Golgi, probably owing to the same reason as for Wipi2. We showed the data of STED analysis of Wipi2 in the revised manuscript (Suppl. Fig. 15a), and added a description to the

Discussion section. as follows. (page 22, lines 3–9)

“Wipi2 is a paralog of Wipi3, with a similar structure, including the PI3P-interacting domain. However, the roles of Wipi2 and Wipi3 are different. The main cause of this difference is thought to be the efficiency of translocation to the Golgi upon a stimulus. We here described the requirement of the PI3P-interacting domain for translocation of Wipi3 to the Golgi. However, this domain is not sufficient for this translocation. Another crucial signal sequence is thought to be required for the translocation of Wipi3 to the Golgi, which may be absent in Wipi2.”

Comment #3: Despite the presence of “neuron” in the title of the paper, very few experiments are actually in neurons.

Response #3:

In accordance with this suggestion, we replaced the title to “Wipi3 is essential for alternative autophagy, and its loss causes neurodegeneration via a mechanism different from canonical autophagy”

Comment #4: There is at least one other Wipi3 antibody that the authors did not mention in the text – from Santa Cruz. Have the authors tried this reagent or considered making their own antibody?

Response #4:

We have tested the following anti-Wipi3 antibodies: Thermo Fisher Scientific (PA5-50864), Santa Cruz Biotechnology (sc-514194), and Invitrogen (PA5-68658, PA5-68659). The antibody PA5-50864 was useful for western blotting, which was performed in this manuscript. However, we could not find any useful antibodies for immunofluorescence (Attached Figure 2). We also generated our own polyclonal antibodies, but they could not be used for immunofluorescence.

Comment #5: For Figure 1e, there are no controls shown (wild type cells, untreated).

Response #5:

In accordance with this comments, we added control images to Figure 1e of the revised manuscript.

Comment #6: For Fig 2J-K, why do the Rab9 channels look so different between NT and Wipi3Cr with etoposide?

Response #6:

We appreciate this important comment. Rab9 usually localizes to the Golgi and late endosomes, and their distribution varies between each individual cell. In the original image of untreated *Atg5^{KO}* MEFs, Rab9 appeared to be mainly localized to the Golgi, whereas in the image of etoposide-treated *Atg5^{KO}/Wipi3^{Cr}* MEFs, Rab9 appeared to be mainly localized to the endosome. However, this difference was simply a result of the inappropriate selection of the images, because we observed small yellow puncta localized to the Golgi and late endosomes in both types of cells. Therefore, we replaced these images to more representative ones in the revised manuscript.

Comment #7: What is the difference between Fig 3D and Fig 3H? If the y-axis graph label in Fig 3H is correct, the graph is very confusing, as there are many puncta in both conditions. If the graph is truly of total Wipi3 puncta, then Fig 3C and Fig S11 are not representative images, as both show many Wipi3 puncta in the NT condition. If the y-axis label in Fig 3H is wrong and should read “Cells with Golgi-localized Wipi3 puncta,” I don’t understand how Fig 3C and Fig 3H are different.

Response #7:

We appreciate this comment and apologize for our confusing description. This confusion was a result of our inappropriate wording; we used the word “puncta” in both the STED imaging (Fig. 3D) and immunofluorescence (Fig. 3H) of the original manuscript. However, because of the difference in resolution between the two methods, the puncta in Fig. 3D and Fig. 3H indicate different objects. To avoid confusion, we deleted the quantitative data from the STED and immunofluorescence images, and we

instead added quantitative data from the Duolink images (Fig. 3e and f) to the revised manuscript. We also used the word “puncta” carefully in the revised manuscript.

Comment #8: For the mRFP-GFP and Kik fluorescent micrographs, nearly all of them are overexposed, making it virtually impossible to determine anything from them. In addition, colocalization between yellow and cyan is impossible to see. Cyan should be changed to a different color.

Response #8: We greatly appreciate this comment. We replaced some of the images to those that are not overexposed. We also used blue color for colocalization with yellow.

Comment #9: More broadly, the Kikume assay seems of limited utility – it is simply a cytosol exclusion assay, and thus not actually specific for the autophagy compartment they are investigating.

Response #9:

To date, we have not observed any merging of Kikume with organelles other than autophagosomes/autolysosomes in our experiments. However, we do not formally deny the possibility that other organelles are also observed as red puncta of Kikume in some situations. Therefore, we removed all the data using the Kikume assay from the revised manuscript.

Comment #10: In Fig. S3A-B, the micrographs are not of the same cell in each panel, which does not inspire confidence in the new alternative autophagy assay.

Response #10:

As described above, we removed all the data obtained using the Kikume assay from the revised manuscript.

Comment #11: In Fig S9B, the n=4. This should be much higher for MEFs. For Fig S9C, why isn't WIPI3Cr tested? Same question for Fig S9.

Response #11: We repeated the experiment shown in Suppl. Fig. 8b (Fig. S9B in the original manuscript) a total of 6 times. We also analyzed the population of apoptotic cells upon etoposide treatment using *Atg5^{KO}/Wipi3^{Cr}* MEFs in addition to *Atg5^{KO}* MEFs, and showed the results in Suppl. Fig. 8c (which was improved from Fig. S9C of the original manuscript).

Comment #12: In line 190, “diffusely” should be changed to “evenly” – the signal is definitely punctate, not diffuse.

Response #12:

In accordance with this comment, we replaced the word “diffusely” to “evenly” in the revised manuscript.

Comment #13: In line 195-196, “some Flag-Wipi3 appeared as cytoplasmic puncta together with the trans-Golgi” doesn’t make sense – is Wipi3 cytoplasmic, punctate, or in the trans-Golgi? I don’t think it can be all three at once. The same problem can be found in the figure legend. The Wipi3 puncta just look like background IF staining more than actual, biologically relevant puncta.

Response #13:

In accordance with these comments, we removed this description. To show more reliable and easily understandable data, we performed the Duolink assay and counted the number of signals to quantify *trans*-Golgi-localized Wipi3.

Comment #14: In line 196, in conjunction with Fig 3I, puncta are referred to as “Wipi3/trans-Golgi puncta,” but there is no Golgi marker in those micrographs. “trans-Golgi” should be removed.

Response #14:

The Fig. 3I of the original manuscript indicates the colocalization of Wipi3 and Kikume puncta. Because we removed all the data from the Kikume assay, we also removed this description. Instead, we showed the clear colocalization of Wipi3 and red puncta of

mRFP-GFP (autolysosomes) in Figure 3g and 6g of the revised manuscript.

Comment #15: For Fig S15F&H, why are the p62 results opposite while the LC3-II results are congruent?

Response #15: We greatly appreciate the reviewer for pointing this out. These were strange results that we obtained, but when we performed the same experiments repeatedly, we obtained reasonable data in every experiments. We hence show our new data in Figure 6d and e of the revise manuscript.

Comment #16: The manuscript could not be fully assessed because data panels were missing for Fig S9D&E, Fig S10, and Fig S16C.

Response #16:

We apologize for this problem. Some data panels may have disappeared during uploading of the manuscript. We will carefully check the uploaded version of the manuscript before resubmission?

Comment #17: In Fig S22D, how do we know that the structures are mitochondria? They don't look like mitochondria to me.

Response #17:

We showed severely damaged mitochondria in Fig S22D of the original manuscript, and hence they did not look like conventional mitochondria. Therefore, we presented mildly damaged mitochondria with remaining cristae in Suppl. Fig. 19d of the revised manuscript.

Comment #18: What are we supposed to get from Fig S23? The two micrographs are very different magnifications and the “structures” that the arrows point to look more like EM artifacts than real, biologically relevant formations.

Response #18:

We appreciate this comment. The photos show the most notable abnormal structure in *Atg7^{CKO}* Purkinje cells and *Wipi3^{CKO}* Purkinje cells, which were an inclusion body and fibrils, respectively. They are not EM artifacts. To avoid misunderstanding, we showed the photos with the same magnification in Suppl. Fig. 21 of the revised manuscript, and describe their importance in the figure legend.

Comment #19: Fig S26B-C should be moved to Fig 6.

Response #19:

In accordance with this comment, we moved Fig. S26, B-C to Fig. 9 in the revised manuscript.

Comment #20: In sum, this work is potentially of interest to the field, but the current manuscript is dense and difficult to read. The work is overly broad and insufficiently deep on some key points. And there are a range of issues, such as those detailed above, that raise concern about the ability of the authors to cleanly present each experiment and judiciously interpret each result in the forest of assays and experimental results reported here.

Response #20:

We greatly appreciate this constructive comment. As proposed, we performed various experiments to fill the gaps in our previous manuscript, reorganized the findings, and extensively rewrote the manuscript so that it is easy to read.

Responses to the comments by Reviewer #3

General Comment: In this manuscript, Yamaguchi et al address the role of WIPI3 in alternative autophagy both in intact cells and in vivo, after generating a neuron-specific conditional KO mouse. The authors begin their studies by focusing on yeast cells, where they have identified an experimental paradigm of noncanonical autophagy by knocking out *Atg5* and inducing the alternative pathway with amphotericin B1. They identified a number of genes operating in the alternative autophagy pathway, including *Hsv2*, which is the ortholog of WIPI3 or 4. They decide to pursue WIPI3 in mammalian cells, because it is the PROPPIN isoform that appears to be the most relevant for alternative autophagy. The authors make a reasonable case for an involvement of WIPI3 in alternative autophagy, by using an *Atg5* null background in combination with etoposide treatment and showing a requirement for this protein in a variety of morphological phenotypes. They finally generate a neuron-specific cKO mouse for WIPI3 and find a variety of phenotypes in the CNS, including loss of Purkinje neurons, astro- and microgliosis, increased presence of fibrils and swollen Golgi in Purkinje cells. A subset of phenotypes can be rescued by overexpressing with lentivirus the protein DRAM1, which is a Golgi protein also involved in alternative autophagy and can rescue WIPI3 loss-of-function phenotypes in cell culture.

Response: We greatly appreciate these constructive comments.

Comment #1: Overall, this is an interesting study that documents a number of unexpected phenotypes in the WIPI3 KO cells in vitro and neurons in vivo. The morphological analyses presented in the study are spectacular, although there isn't a systematic morphometric analysis of key findings, making it difficult to assess how common or prevalent the phenotypes are.

Response #1:

We appreciate this important comment. In accordance with this comment, we summarized the morphological features of *Wipi3^{KO}*, *Wipi3^{Cr}*, *Atg5^{KO}/Wipi3^{Cr}*, and *Atg7^{KO}/Wipi3^{KO}* MEFs and *Wipi3^{cKO}* and *Atg7^{cKO}/Wipi3^{cKO}* neuronal cells. In all the cells,

abnormal Golgi morphologies were observed, in which Golgi membranes were ministacked, swollen, and rod-shaped. Therefore, Wipi3 is thought to affect Golgi membranes. We included this description in the Discussion section of the revised manuscript, as follows. (page 21, lines 9–23)

“We here analyzed the ultrastructural morphology of *Wipi3^{KO}*, *Wipi3^{Cr}*, *Atg5^{KO}/Wipi3^{Cr}*, and *Atg7^{KO}/Wipi3^{KO}* MEFs and *Wipi3^{cKO}* and *Atg7^{cKO}/Wipi3^{cKO}* neuronal cells. The most notable and common alteration among these cells occurred in the Golgi, in which Golgi membranes were ministacked, swollen, and rod-shaped. Such alterations in Golgi membranes were not observed in *Atg5^{KO}* MEFs and *Atg7^{cKO}* neuronal cells, and have also not been reported in numerous EM analyses of canonical autophagy-deficient cells. Therefore, Wipi3 is thought to affect to the Golgi membranes in a manner different from that of canonical autophagy. The morphologies of MEFs and neurons lacking Wipi3 were basically the same, but the accumulation of fibrils was only observed in Wipi3-deficient neurons. This is thought to be owing to differences in the cell types or the time required for of accumulation; i.e., many substrates are known to take a long time to accumulate in neurons. The loss of both Wipi3 and Atg5 or Atg7 additionally abolished a small fraction of autophagic structures from MEFs and neurons, and induced the accumulation of abnormal smooth ER in neurons, which was owing to the blockage of canonical autophagy.”

Comment #2: However, while the authors can clearly show a role for WIPI3 in alternative autophagy in vitro using artificial conditions, such as lack of key components of the classical autophagy (e.g. ATG5 or 7) combined with etoposide treatment, there is no evidence that specific defects in alternative autophagy are what accounts for the phenotypes in neurons in vivo.

Response #2:

This is an important point. From a morphological aspect, alterations in Golgi morphology were commonly observed in all Wipi3-deficient cells, including neuronal cells. Furthermore, this abnormal morphology was rescued by the expression of *Dram1*, another molecule involved in alternative autophagy. Therefore, abnormal Golgi morphology appears to be mediated by the loss of alternative autophagy.

Furthermore, we newly found the accumulation of ceruloplasmin, which is a cargo molecule of alternative autophagy, in *Wipi3^{CKO}* brain cells. This accumulation is thought to be caused by defects in alternative autophagy, and may be a cause of the neurodegeneration.

Comment #3: Both WIPI3 and DRAM1 are Golgi-associated proteins and lack of WIPI3 causes a Golgi phenotype in Purkinje cells which can be rescued with DRAM1. These findings do not prove that these proteins have any connections with alternative autophagy. In fact, it may simply show that messing up with the Golgi is deleterious for neurons. The only piece of evidence suggestive of a role of WIPI3 in some form of autophagy is the accumulation of fibrils in the KO Purkinje cells. However, there is no information on what those fibrillar structures may be and thus no direct way to test whether these structures arise from defects in alternative autophagy in vitro or in vivo. In the absence of this kind of evidence, the conclusions of this manuscript are largely overstated and the paper is more suitable for a specialized journal, after toning down the interpretations of the in vivo work. Unfortunately, the physiological role of alternative autophagy is not clarified by this study.

Response #3:

We believe these are very important comments (similar points were raised by Reviewer #1). To understand the role of *Wipi3* and alternative autophagy, it is useful to identify cargos that are accumulated in *Wipi3*-deficient cells and mice. The reviewer suggested that we should identify the component of fibrils accumulated in the *Wipi3^{CKO}* brains, and we extensively analyzed fibrils by their isolation followed by mass spectrometry. However, we were unable to identify its main components.

On the other hand, we found iron deposition and the accumulation of ceruloplasmin in *Wipi3^{CKO}* brains. Ceruloplasmin is a ferroxidase, and hence its accumulation accounts for the iron deposition. Because ceruloplasmin is degraded by *Wipi3*-dependent alternative autophagy *in vitro*, the accumulation of ceruloplasmin in *Wipi3^{CKO}* brains is thought to be caused by a failure of *Wipi3*-dependent alternative autophagy. This may account for the iron deposition in *Wipi3^{CKO}* brains, and may be one cause of the

neurodegenerative disorder observed in *Wipi3^{CKO}* mice.

Comment #4: - Lines 24-26: please be more specific and state what is different about the morphology, rather than stating that they are different.

Response #4:

In accordance with this comment, we replaced our description to the following in the revised manuscript. (page 2, lines 10–13)

“Although *Atg7*-deficient mice showed similar phenotypes to *Wipi3*-deficient mice, electron microscopic analysis showed that they have completely different subcellular morphologies, including the morphology of organelles.”

Comment #5: - Lines 39-40: The statement “Recently, this machinery was reported to work slowly even in the absence of *Atg5*” seems odd, as the autophagy machinery is precisely expected to work slowly in the absence of *Atg5*.

Response #5:

According to this comment, we replaced our description to the following in the revised manuscript. (page 3, lines 13–15)

“Previously, this mechanism was considered to work in a manner entirely dependent on *Atg5*, but recent observations have suggested that it functions even in the absence of *Atg5*”

Comment #6: - Line 68: “is blocked” rather than “are blocked”.

Response #6:

Thank you very much for pointing out our mistake. We replaced “are blocked” to “is blocked” in the revised manuscript.

Attached Fig. 1. *Wipi3* deficiency has little effect on nucleophagy

(a, b) The indicated MEFs were left untreated (NT) or treated with etoposide (10 μ M). We also added E64d and pepstatin to visualize cargos in autolysosomes. After 18 hr, cells were stained with anti-Histon-H1 (linker histone) or anti-nucleolin (ubiquitously distributed protein in the nucleolus) antibodies together with anti-Lamp2 antibody. Large Lamp2 puncta (autolysosomes) were observed in *Atg5^{KO}* MEFs, but not *Atg5^{KO}/Wipi3^{Cr}* MEFs. We did not observe engulfed Histon-H1 and nucleolin into autolysosomes. Nuclear morphology was also appeared normal. (c) The indicated MEFs were left untreated (NT) or treated with etoposide (10 μ M). After 18 hr, cells were stained with anti-dsDNA antibody and anti-Lamp2 antibodies. We did not observe any differences between these MEFs. (d) The indicated cerebellum were stained with the indicated antibodies. Nuclei were looked normal, and we did not observe any differences between these mice.

Attached Fig. 2. No useful antibodies for *Wipi3* immunofluorescence.

The indicated MEFs were immunostained with the indicated anti-*Wipi3* antibodies. However, we did not find any difference between WT MEF and *Wipi3^{Cr}* MEFs.

Attached Fig. 1 Yamaguchi et al.

a

b

c

d

Attached Fig. 2 Yamaguchi et al.

REVIEWERS' COMMENTS

Reviewer #1 (Remarks to the Author):

The revised version of this manuscript has improved readability and has streamlined some of its content.

Given that the authors did not wish to split the manuscript the removal of the whole Kikume part has helped focusing the work.

The authors have addressed all my previous points and removal of Kikume solved others, therefore I am inclined to recommend for publication.

Some minor modifications that would further help the manuscript structure are

- the results in the associated figure to the rebuttal could be mentioned in the manuscript as data not shown as they would be of relevance to the readers
- the basically negative results re Wipi3 function in canonical autophagy in Figure 6 could be better placed in the supplementary and actually they could be the very first dataset to justify the focus of this work on alternative autophagy and Wipi3 in particular.

Reviewer #2 (Remarks to the Author):

The manuscript by Yamaguchi et al. describes interesting, and potentially very important new findings on the role of alternative autophagy. The work describes a broad array of findings, using many different assays and approaches. In the initial version of this work, the manuscript was very difficult to read, and the authors have significantly improved the clarity of their presentation. More importantly, they have now filled in some of the gaps identified in the previous round of review. Some of the problems of identified in the initial submission remain - the work is too broad and not always sufficiently deep. There is too much here for a single paper, and not all points the authors wish to make are fully convincing. However, there is no doubt that the work has improved very significantly, and that the observations discussed will be of great interest to many.

Based on both the importance of the findings and the improvements the authors have made in revision, I recommend publication.

Reviewer #3 (Remarks to the Author):

The authors have addressed all my comments. The addition of new data, including ceruloplasmin data, significantly strengthens the manuscript and makes a strong case for defects in non-canonical autophagy driving the neurodegenerative phenotypes of the WIPI3 KO mouse. It may also have major relevance for WIPI-related neurodegenerative disorders associated with brain iron accumulation.

Point-by-point Responses to the Editor's and Reviewer's comments

We therefore invite you to revise your paper one last time to address the remaining concerns of our reviewers. Notably, we would require the data currently in the rebuttal to be included in the article as either a Supplementary or Main Figure (as requested by Reviewer #1 – though not as “Data not shown”). However, we would not necessarily require a rearrangement of the Figure 6 (as otherwise suggested by Reviewer #1).

Response:

In accordance with this suggestion, we included the “Attached Figures for Reviewers” in the article as Supplementary Figures 10 and 23, and described them as follows.

" there are no available Wipi3 antibodies that can be used for immunofluorescence analyses (Suppl. Fig.10)"

" However, analysis of DNA and nuclear factors did not show any apparent differences between Wipi3-expressing cells and Wipi3-deficient cells (Suppl. Fig. 23)."

At the same time we ask that you edit your manuscript to comply with our policies and formatting requirements and to maximise the accessibility and therefore the impact of your work.

Response:

In accordance with this comment, we have responded and described in author checklist.

We all thank the editor for helpful suggestions.